# Regional Influence of Wildfires on Aerosol Chemistry in the Western US and Insights into Atmospheric Aging of Biomass Burning Organic Aerosol

Shan Zhou[1], Sonya Collier[1], Daniel A. Jaffe[2,3], Nicole L. Briggs[2,3,4], Jonathan Hee[2,3], Arthur J. Sedlacek III[5], Lawrence Kleinman[5], Timothy B. Onasch[6], Qi Zhang[1*]

[1]Department of Environmental Toxicology, University of California, Davis, CA 95616, USA
[2]School of Science, Technology, Engineering, and Mathematics, University of Washington Bothell, Bothell, WA 98011, USA
[3]Department of Atmospheric Sciences, University of Washington, Seattle, WA 98195, USA
[4]Gradient, Seattle, WA 98101, USA
[5]Environmental and Climate Sciences Department, Brookhaven National Laboratory, Upton, NY 11973, USA
[6]Aerodyne Research Inc., Billerica, MA 01821, USA

*Corresponding Author*: Qi Zhang (dkwzhang@ucdavis.edu), (530)752-5779, Department of Environmental Toxicology, University of California, Davis, CA 95616

**Abstract.**

Biomass burning (BB) is one of the most important contributors to atmospheric aerosols on a global scale and wildfires are a large source of emissions that impact regional air quality and global climate. As part of the Biomass Burning Observation Project (BBOP) field campaign in summer 2013, we deployed a High Resolution Time-of-Flight Aerosol Mass Spectrometer (HR-AMS) coupled with a thermodenuder at the Mt. Bachelor Observatory (MBO, ~2.8 km above sea level) to characterize the impact of wildfire emissions on aerosol loading and properties in the Pacific Northwest region of the United States. MBO represents a remote background site in the western U.S. and it is frequently influenced by transported wildfire plumes during summer. Very clean conditions were observed at this site during periods without BB influence where the 5-min average ($\pm$ 1$\sigma$) concentration of non-refractory submicron aerosols (NR-PM$_1$) was 3.7 $\pm$ 4.2 $\mu$g m$^{-3}$. Aerosol concentration increased substantially (reaching up to 210 $\mu$g m$^{-3}$ of NR-PM$_1$) for periods impacted by transported BB plumes and aerosol composition was overwhelmingly organic. Based on Positive Matrix Factorization (PMF) of the HR-AMS data, three types of BB organic aerosol (BBOA) were identified, including a fresh, semivolatile BBOA-1 (O/C = 0.35; 20% of OA mass) that correlated well with ammonium nitrate, an intermediately oxidized BBOA-2 (O/C = 0.60; 17% of OA mass), and a highly oxidized BBOA-3 (O/C = 1.06; 31% of OA mass) that showed very low volatility with only ~ 40% mass loss at 200$^{\circ}$C. The remaining 32% of the organic aerosol (OA) mass was attributed to a boundary layer (BL) oxygenated OA (BL-OOA; O/C = 0.69) representing OA influenced by BL dynamics and a low-volatility oxygenated OA (LV-OOA; O/C = 1.09) representing regional free troposphere aerosol. The mass spectrum of BBOA-3 resembled that of LV-OOA and had negligible contributions from the HR-AMS BB tracer ions – C$_2$H$_4$O$_2^+$ (*m/z* = 60.021) and C$_3$H$_5$O$_2^+$ (*m/z* = 73.029). This finding highlights the possibility that the influence of BB emission could be underestimated in regional air masses where highly oxidized BBOA (e.g., BBOA-3) might be a significant

aerosol component. We also examined OA chemical evolution for persistent BB plume events originating from a single fire source and found that longer solar radiation led to higher mass fraction of the chemically aged BBOA-2 and BBOA-3 and more oxidized aerosol. However, an analysis of the enhancement ratios of OA relative to CO ($\Delta OA/\Delta CO$) showed little difference between BB plumes transported primarily at night versus during the day, despite evidence of substantial chemical transformation in OA induced by photo-oxidation. These results indicate negligible net OA production in photochemically-aged wildfire plumes observed in this study, for which a possible reason is that SOA formation was almost entirely balanced by BBOA volatilization.

## 1   Introduction

Biomass burning (BB) is estimated to be the largest source of primary carbonaceous aerosols and a major source of reactive trace gases in the Earth's atmosphere (Bond et al., 2004; Akagi et al., 2011). Emissions from wildfires and other BB sources, such as residential wood combustion and agricultural burning, have been shown to affect the global radiation budget (IPCC, 2013) and degrade air quality in both rural areas and populated locations (e.g. Jaffe et al., 2008; Jaffe and Wigder, 2012). The environmental impacts of BB emissions are strongly correlated with the chemical, optical, and microphysical properties of BB aerosols, which are in turn dependent in a complex manner on fuel type, combustion phase, and atmospheric aging of emitted particles and gas species (e.g., Petters et al., 2009; Liu et al., 2014; Collier et al., 2016).

Organic compounds are a dominant component of BB aerosols (Bond et al., 2004; De Gouw and Jimenez, 2009), but the chemical and physical properties of primary organic aerosol (POA) released directly from burning and secondary organic aerosol (SOA) formed from gaseous precursors emitted by BB are dramatically different. For example, BB POA tends to be semivolatile, smaller in size, and composed of less oxidized compounds, whereas SOA from BB is generally more oxidized, larger in size, and less volatile (Abel et al., 2003; Heringa et al., 2011; May et al., 2013). Furthermore, aerosol composition, optical properties, and hygroscopicity have been found to change substantially in BB plumes undergoing photo-oxidation and cloud processing and the changes are mostly driven by the organic fraction (Abel et al., 2003; de Gouw et al., 2006; Engelhart et al., 2012; Gilardoni et al., 2016). Understanding the chemical properties and atmospheric processing of organic aerosols (OA) from BB sources (i.e., BBOA) is thus crucial for improving our ability to quantitatively assess and predict the impacts of BB emissions on climate and air quality. However, the chemical processing of BBOA is highly complex and the net effect of aging on BBOA mass is highly variable. For example, while several laboratory studies reported substantial formation of SOA during chamber aging, others observed a very small increase or even a decrease of BBOA mass (Grieshop et al., 2009; Cubison et al., 2011; Hennigan et al., 2011; Heringa et al., 2011; Ortega et al., 2013). Field studies have also observed enhancement (Yokelson et al., 2009; DeCarlo et al., 2010), depletion (e.g.,Akagi et al., 2012; Jolleys et al., 2015), or no change (Brito et al., 2014; May et al., 2015), of dilution-adjusted OA mass in BB plumes after emissions.

In order to decipher what factors affect BBOA evolution and reconcile discrepancies in previous laboratory and atmospheric observational results, the U.S. Department of Energy (DOE) sponsored the Biomass Burning Observation Project (BBOP) campaign, which combined aircraft-based measurements with mountain top

observations to characterize the downstream evolution of the chemical, microphysical, and optical properties of
carbonaceous aerosol generated by BB. Wildfires across the western U.S. have been linked to increased $PM_{2.5}$
concentrations at various receptor sites (Jaffe et al., 2008) and high pollution episodes that exceeded the National
Ambient Air Quality Standards (Jaffe and Wigder, 2012). Furthermore, due to changes in precipitation, temperature
and other meteorological conditions as a result of climate change, wildfire activities in this region have been
increasing (Westerling et al., 2006; Dennison et al., 2014) and are predicted to increase summertime OA
concentration by 40% from 2000 - 2050 (Spracklen et al., 2009).
A large number of wildfire events originating in the western US were observed during BBOP from the Mount
Bachelor Observatory (MBO) – a remote mountain-top site that serves to characterize western U.S. background
conditions and is frequently impacted by transported BB plumes during the summer fire season (Wigder et al., 2013).
Continuous measurements of BB plumes at MBO allow for the study of BBOA with different source, age, and
formation pathways under realistic atmospheric conditions and can provide rich data for evaluating the impact of BB
emissions on regional aerosol properties and elucidating their atmospheric aging processes. A number of recent
studies were conducted at fixed locations in the western U.S. and investigated impacts of BB on ozone, gaseous
nitrogen species, and organic and elemental carbon (e.g.,Wigder et al., 2013; Timonen et al., 2014; Hallar et al.,
2015). Yet, only a few ground-based measurements have examined the chemical composition and evolution of
BBOA, including a filter-based study of wildfire aerosols in Yosemite National Park (Engling et al., 2006 ) and a
single-particle mass spectrometry study on the mixing state and aging of particles during the 2007 San Diego
wildfires (Zauscher et al., 2013).
In this study, we provide an overview of the chemical and physical characteristics of non-refractory
submicrometer particles (NR-$PM_1$) at MBO and examine the changes in ambient aerosol concentration and
composition influenced by BB emissions. The sources of OA are investigated via factor analysis of the HR-AMS
data and the aging of BBOA are discussed via combining real-time measurements with trajectory analysis. We also
examine the enhancement and chemical transformation of OA in BB plumes transported during day-time and night-
time, respectively.

## 2 Experimental methods

### 2.1 Sampling site and wildfires in the vicinity

The Mt. Bachelor Observatory (43.981°N 121.691°W, Fig. 1) is situated on the summit of Mt. Bachelor (~ 2.8
km a.s.l.), an isolated volcanic peak in the Deschutes National (coniferous) Forest in central Oregon. The nearest
populated areas are Bend (pop. ~80,000), 31 km to the east, and Redmond (pop. ~ 26,000), 53 km northeast of MBO.
Due to its high elevation and distance from local pollution sources, MBO is a remote background site in the western
U.S. well positioned for sampling of background free tropospheric air and observation of long-range transport of
Asian plumes and North American wildfires (Weiss-Penzias et al., 2006; Wigder et al., 2013; Briggs et al., 2016).
During the sampling period from July 25 to August 25, 2013, various active wildfires in northern California and
southeastern and central Oregon were detected by the Moderate Resolution Imaging Spectroradiometer (MODIS)

 satellite (https://firms.modaps.eosdis.nasa.gov) (Fig. 1). Three intense fires, the Salmon River Complex Fire (SRCF),

Whiskey Complex Fire (WCF) and Douglas Complex Fire (DCF), were active for a majority of the time during this
study and hence were identified as major fires in the region.

## 2.2 Real-time measurements at MBO

Continuous observations at MBO included submicron aerosol light scattering (TSI nephelometer; 450, 550, and
700 nm) and absorption (Radiance Research PSAP; 467, 530, and 660 nm), elemental and organic carbon (Sunset
Lab), CO and $CO_2$ (Picarro Cavity Ring-Down Spectroscopy), $O_3$ (Dasibi), $NO_x$ (Air Quality Design 2-channel
chemiluminescence), $NO_y$ (chemiluminescence), peroxyacetyl nitrate (PAN; custom gas chromatograph), and
meteorological parameters (e.g., Weiss-Penzias et al., 2006; Briggs et al., 2016). Data reported in this study are
5-min averages. During this study, an HR-AMS (DeCarlo et al., 2006) was deployed downstream of a
thermodenuder (TD) to measure the size-resolved composition and volatility of NR-PM$_1$. These are the first real-
time aerosol chemical measurements at MBO. The TD consists of a heated tube followed by a heated adsorption
section that uses carbon cloth to prevent recondensation of organic vapors (Fierz et al., 2007). The TD was
automated using a custom program to step through 12 different temperatures ranging from 30 to 200°C, at 10 min
time intervals. Changes in mass and chemical composition of NR-PM$_1$ as a result of aerosol evaporation were
quantified by the HR-AMS by alternating sampling between the TD and the bypass (BP) ambient sampling mode
every 5 min. During BP mode, the temperature in the heated section ramped up to the next setting and reached
thermal stability before switching back to TD mode. The switching between sampling modes was triggered by a
digital output signal from the HR-AMS which was synchronized to the HR-AMS averaging intervals and was
achieved using an actuated 3-way ball valve. Aerosol residence time in the TD was 8.2 s at the experimental flow
rate (1.1 L min$^{-1}$). Particle losses within the TD mode (~ 5%) due to diffusional and thermophoretic forces were
quantified based on the behavior of ammonium sulfate.

## 2.3 HR-AMS data analysis

The HR-AMS was operated in the ion optical "V-mode" with reduced micro-channel plate bandwidth due to
signal interference at MBO, and was calibrated following standard protocols described in detail in Collier et al.
(2016). Data analyses were performed utilizing AMS analysis toolkit SQUIRREL v1.53 and PIKA v1.12 in Igor Pro
6.34A (Wavemetrics, Inc., Lake Oswego, OR). Default relative ionization efficiency (RIE) values were assumed for
organics (1.4), nitrate (1.1), and chloride (1.3), while an RIE value of 5 was determined for ammonium and 1.32 for
sulfate following the analysis of pure $NH_4NO_3$ and $(NH_4)_2SO_4$, respectively. A time- and composition-dependent
collection efficiency (CE) was applied based on the algorithm by Middlebrook et al. (2012), leading to an average
(±1σ) CE of 0.56 (±0.12). Time-dependent gas phase $CO_2^+$ subtraction was performed to improve the determination
of OA, which is critical for low OA concentration periods (Collier and Zhang, 2013). The mass concentrations of
ammonium, nitrate, chloride, and sulfate were determined from PIKA analysis of the high-resolution mass spectra
(HRMS) whereas organic concentrations came from SQUIRREL analysis of the unit mass resolution (UMR) data
after adjusting the fragmentation table (Allan et al., 2004) to properly represent the organic signals at *m/z*'s that are
contributed significantly by inorganic or air signals. The detection limits of organics, sulfate, nitrate, ammonium,
and chloride, defined as 3 times the standard deviations (3σ) of the corresponding signals in particle-free ambient air,
were 28.1, 4.5, 2.3, 9.6, and 3.0 ng m$^{-3}$, respectively, for an averaging time of 5 min. Atomic oxygen-to-carbon (O/C)
and hydrogen-to-carbon (H/C) ratios and the organic mass-to-carbon (OM/OC) ratio were determined using the
Improved-Ambient (IA) method (Canagaratna et al., 2015). We also reported the ratios determined using the
previously published Aiken-Ambient (AA) method (Aiken et al., 2008) in order to compare with literature results.
As shown in Fig. S1 in the Supplement, the O/C, H/C, and OM/OC values determined from the two methods
correlate tightly ($r^2$ = 0.99), and the IA method reports 29%, 5%, and 31%, respectively, higher values compared to
the AA method.

152         Positive Matrix Factorization (PMF) was executed using the PMF2 algorithm (Paatero and Tapper, 1994) in the

PET v2.05 program (Ulbrich et al., 2009). The spectral matrices of organic and inorganic species were combined
(Sun et al., 2012) and the ion signals were expressed in nitrate-equivalent concentrations. Periods with organic
concentration below 1.5 μg m$^{-3}$ (~ 20% of the total data points), which hindered the model to converge due to
increased rotational ambiguity, were excluded from PMF analysis. The HRMS of organic ions at $m/z$ 12 – 180 and
the UMR signals at $m/z$ 181 – 350 were included. For inorganics, only the major ions for each species were included,
i.e., $SO^+$, $SO_2^+$, $HSO_2^+$, $SO_3^+$, $HSO_3^+$, and $H_2SO_4^+$ for sulfate, $NO^+$ and $NO_2^+$ for nitrate, $NH^+$, $NH_2^+$, and $NH_3^+$ for
ammonium, and $HCl^+$ for chloride. $Cl^+$ was not included due to low S/N. Data preparation prior to PMF analysis
followed the steps outlined in the Table 1 of Zhang et al. (2011). After PMF analysis, the mass concentration of each
OA factor was derived from the sum of organic signals in the corresponding mass spectrum after applying the RIE
(=1.4) for organics and the time-dependent CE determined based on aerosol composition (see previous discussion).
The solutions for 3 to 8 factors were explored with varying rotational parameters (-0.5 ≤ FPEAK ≤ 0.5, in
increments of 0.1). After a detailed evaluation of mass spectral profiles, temporal trends, diurnal variations, and
correlations with external tracers, the five-factor solution with FPEAK = 0 was chosen. The diagnostic information
for five-factor solution is shown in Fig. S2. In comparison, the four-factor solution resulted in large residual signals,
indicating that an additional factor was needed to explain the variation in the data, whereas the six-factor solution
showed indications of factor splitting, suggesting that too many factors were introduced (Fig. S3). PMF was also
performed on the organic spectra only but wasn't able to resolve two types of OOA (OA (see more detailed
discussions in Section 1 of the Supplement).

171         The concentrations of OA factors at different TD temperatures were determined via multivariate linear regression

of the HRMS of OA after TD against the HRMS of the 5 OA factors determined from PMF of the ambient OA data
following the procedures given in Zhou et al. (2016). The mass fraction remaining (MFR) of a factor at each TD
temperature was then determined as the slope from orthogonal fit between the time series after TD and the ambient
time series. The mass fraction remaining (MFR) of a factor at each TD temperature was then determined as the slope
from orthogonal fit between the time series after TD and the ambient time series. Note that the uncertainties of the
MFR likely increase with TD temperature, especially for more volatile species, possibly due to changes in particle
collection efficiency and decreased concentration (thus lower S/N). Indeed, as shown in Fig. S4, the correlation
coefficients between the TD-processed aerosol species and the ambient data decreased with increased TD
temperature. Thermograms, which describe the MFR as a function of temperature, have been corrected for particle
losses in the TD mode. Aerosol data reported here have all been converted to concentrations at standard temperature
and pressure (STP, 273 K, 1 atmosphere).

**2.4 Back trajectory analysis and calculations of plume transport time and cumulative solar radiation**

The HYbrid Single Particle Lagrangian Integrated Trajectory (HYSPLIT) model backward air mass trajectories
(Draxler, 1998) were initiated from MBO at one-hour intervals throughout the campaign period. Three-day
backward trajectories using the 40 km resolution US Eta Data Assimilation System (EDAS) meteorological data
(http://ready.arl.noaa.gov/HYSPLIT.php) were calculated at a starting height of 1500m above ground level.
Meteorological variables (e.g. solar radiation and relative humidity (RH)) along the trajectories were also model
outputs. By overlapping the back trajectories with MODIS fire hotspots, we estimated the transport times for BB
plumes that unambiguously passed over active fire sources (Collier et al., 2016). In addition, we also estimated the
cumulative solar radiation exposure and average RH for these plumes during the period between emission at fire
source and arrival at MBO.

**3 Results and discussions**

**3.1 Observations of wildfire-influenced air masses at MBO**

Fig. 2 provides an overview of the meteorological conditions, trace gases mixing ratios, and aerosol
concentration and composition during the sampling period (July 25 – August 25, 2013). The summit air was cool
(average temperature of $11.2 \pm 4$ °C) and dry (average RH of $46 \pm 21\%$), although there were periods (e.g., August
16 and August 23) when MBO was in low clouds and measured RH reached 98%. Wind was generally strong
(average $= 5.7 \pm 3.4$ m s$^{-1}$) with a dominant flow from the west and southwest direction, which provides suitable
conditions for long-range transport of fire smoke from Northern California and Southwest Oregon. Indeed, the
bivariate polar plots of total NR-PM$_1$, submicrometer aerosol light scattering at 550 nm ($\sigma_{550nm}$), and CO (Fig. 1b, 1d,
and 1e) calculated using the OpenAir software (Carslaw and Ropkins, 2012) all show the highest values at a wind
speed of ~ 13 m s$^{-1}$ from the southwest direction, where the major complex fires were located (Fig. 1a).
The average NR-PM$_1$ concentration during the entire sampling period was 15.1 μg m$^{-3}$ and 93% of it was
contributed by organics (Fig. 1c). However, aerosol concentrations and composition changed dynamically. Clean
periods of low concentrations of aerosol (NR-PM$_1$ < 10 μg/m$^3$) and gas-phase pollutants (e.g., CO, NO$_y$, and PAN)
were observed for the first week of sampling (July 25 – 30) and during August 18 – 21 (Fig. 2d – 2f). During these
periods, ammonium sulfate contributed up to 90% of the NR-PM$_1$ mass (Fig. 2g) and the OA spectra showed low
abundances of $C_2H_4O_2^+$ (m/z = 60.021) and $C_3H_5O_2^+$ (m/z = 73.029), which are ion fragments of anhydrous sugar
(e.g., levoglucosan) and HR-AMS tracers for BB (Alfarra et al., 2007). f$_{60}$, which is defined as the fraction of the
signal at m/z 60 (mostly $C_2H_4O_2^+$) in OA spectrum, was generally below 0.3% (Fig. 2h), indicating minimal BB
influence during "clean" periods (Cubison et al., 2011).
In contrast, the other periods were characterized by higher $f_{60}$ (up to 2%), elevated NR-PM$_1$ concentration (up to
$\sim$ 210 µg m$^{-3}$), and larger OA fraction (generally > 90% of NR-PM$_1$; Fig. 2e – 2g). In addition, $\sigma_{550nm}$ (up to $\sim$ 670
Mm$^{-1}$), CO (up to $\sim$ 700 ppbv), NO$_y$ (up to $\sim$ 6.5 ppbv), and PAN (up to $\sim$ 2.2 ppbv) all increased dramatically
during high $f_{60}$ periods (Fig. 2d – 2e). In fact, the time series of all these parameters correlated tightly, with
Pearson's $r^2$ in the range of 0.66 – 0.94 (Fig. S5). These observations highlight the frequent and significant impacts
of wildfire emissions on air quality and atmospheric chemistry in the Pacific Northwest region during this study.
Note that potassium (K) is frequently used as a tracer for BB aerosol and the presence of K in aerosol particles was
clearly observed during high loading periods. However, K concentration in aerosol was overall very low and noisy
throughout this study (Fig. S6), indicating low K contents in wildfire emissions in the western US. Similarly,
Maudlin et al. (2015) observed no strong enhancement of K in wildfire smokes originated from California and
Oregon and concluded that it is not a reliable tracer for BB in this region.

## 224    3.2   Impacts of wildfires on regional aerosol characteristics

### 225    3.2.1. Changes of aerosol concentration and composition due to wildfires

Using $f_{60}$ as an index for the influence of BB emissions on OA composition, we divided the entire campaign into
three regimes: (1) "No BB" for periods with negligible BB influence and $f_{60} \leq 0.3\%$; (2) "BB Infl" for periods with
detectable BB influences and moderately elevated $f_{60}$ values (0.3% - 0.5%); and (3) "BB Plm" for periods with $f_{60} >$
0.5%, indicating intense and less processed BB events. Note that periods with very low OA concentrations (< 1
µg/m$^3$), e.g., August 18 – 21, were classified as "No BB" regardless of the nominal $f_{60}$ values. The average ($\pm$ 1$\sigma$) $f_{60}$
values were 0.18 $\pm$ 0.10%, 0.43 $\pm$ 0.05% and 0.77 $\pm$ 0.29% for "No BB", "BB Infl", and "BB Plm" periods,
respectively (Fig. 3 and Table S1). Similarly, the average mixing ratios of CO, a gaseous pollutant released from
combustion, increased from 87.8 $\pm$ 17.9 ppbv during "No BB" to 121.4 $\pm$ 24.8 ppbv during "BB Infl" and 178.3 $\pm$
68.8 ppbv during "BB Plm" periods.
Fig. 3 shows the comparisons of gas and particle phase properties among the three regimes to illustrate the strong
effects that wildfires have on gases and aerosol composition in the Pacific Northwest region. For example, the
average NR-PM$_1$ concentration was only 3.7 ($\pm$ 4.2) µg m$^{-3}$ during "No BB" but increased by $\sim$ 4 and $\sim$ 7 times,
respectively, during "BB Infl" (13.4 $\pm$ 7.1 µg m$^{-3}$) and "BB Plm" (25.7 $\pm$ 19.9 µg/m$^3$) periods. Aerosol measured at
MBO during "BB Plm" periods was predominantly organic (94.6% of NR-PM$_1$ mass; Fig. S7c). The fraction of OA
in BB aerosols may be fuel dependent, for instance, high values have been reported for ponderosa pine smoke
emissions (99%) (Lewis et al., 2009) and somewhat lower values have been reported for forest fires in south-western
Amazon (93%) (Artaxo et al., 2013) and North America boreal forests (87%) (Kondo et al., 2011), and agricultural
fires in west Africa (85%) (Capes et al., 2008). Even lower values were observed in eastern Mediterranean wildfires
(51.4%) (Bougiatioti et al., 2014) and Asian fires (60%) (Kondo et al., 2011). Since temperate evergreen vegetation
was likely the dominant fuel during this campaign, the high OA/PM$_1$ ratio observed in this study appears consistent
with those of ponderosa pine.
In addition to OA, concentrations of nitrate, ammonium, and chloride all showed substantial increases that
correlated with wildfire impacts (Figs. 2-3, Fig. S8, and Table S1). Nitrate, in particular, displayed large temporal
variations that correlated with wildfire plume influences and its concentration in the "BB Plm" regime was on
average ~ 11 times greater than the "No BB" regime. Nitrate appeared to be bulk neutralized based on comparing
the total molar equivalent of inorganic anions (i.e., sulfate, nitrate and chloride) to that of ammonium (Zhang et al.,
2005) during wildfire-influenced periods (Fig. S9a) and the signal ratios of $NO^+$ to $NO_2^+$ observed in particles
during these periods (2.15 ± 0.006) were very similar to the ratio measured for pure $NH_4NO_3$ particles (2.2; Fig.
S9b), indicating that nitrate was mostly in the form of $NH_4NO_3$. Note that for high organic loading (> 50 μg m$^{-3}$)
periods, excess ammonium relative to sulfate, nitrate, and chloride was frequently observed. A possible reason is the
presence of significant amounts of organic anions in aerosol. Indeed, $CO_2^+$ (*m/z* = 44) and $CHO_2^+$ (*m/z* = 45) – ion
fragments for carboxylic acids – were found to dominate the HRMS of aerosol during periods of high OA loading
(Fig. S7f). Another possible reason is overestimation of ammonium concentration. Biomass burning can emit
significant amounts of nitrogen-containing organic compounds, including amines. These compounds can produce
$NH_x^+$ ions in the AMS, although generally produce significantly more $C_xH_yN^+$ ions (Ge et al., 2014). Tight
correlations between $C_xH_yN^+$ ions and biomass burning tracers (e.g., CO, $C_2H_4O_2^+$, and $C_3H_5O_2^+$) were observed,
suggesting that amino compounds were likely emitted from wildfires in the western US. However, the low
abundance of $C_xH_yN^+$ (~ 0.3% of total organic signal) indicates that organic nitrogen compounds unlikely had a
noticeable influence on ammonium quantification during this study. Sulfate, on the other hand, displayed milder
temporal variation with poor correlation with BB tracers (Fig. 2d-f), indicating that forest fires in this region are not
a significant source of sulfate aerosol. Collier et al. (2016) came to a similar conclusion through examination of
aerosol enhancement ratios in transported BB plumes.
Significant enhancements due to wildfires emissions were also observed for PAN and $NO_y$ (Fig. 3). However, the
mixing ratios of $NO_x$ (mostly as $NO_2$) were comparable among the three regimes. As a result, the fractional
contributions of PAN and particulate nitrate to total $NO_y$ both increased due to wildfire influence (Fig. S10).
Considering that MBO was hours downwind of wildfire sources during this study, this observation is consistent with
the findings of Akagi et al. (2012) that $NO_x$ emitted from BB is rapidly converted to PAN and particulate nitrate
during plume transport, which reflects high levels of acetaldehyde in fire plumes (Akagi et al 2011). The influence
of wildfire emissions on $O_3$ at MBO appears to be complex (Fig. 2c). The average $O_3$ mixing ratio in both "BB Infl"
(49.1 ppbv) and "BB Plm" (47.3 ppbv) regimes were higher than during the "No BB" (44.7 ppbv) periods (Fig. 3),
suggesting that $O_3$ production was enhanced in BB emissions. Similar observations were made previously, which
indicate that $O_3$ tends to peak downwind of fire sources as a result of the interplay of fire emissions (precursors and
reactants) and chemical reactions (Jaffe and Wigder, 2012; Wigder et al., 2013; Briggs et al., 2016).
**3.2.2. Influence of wildfires on organic aerosol chemical properties**
In order to demonstrate the influence of wildfires on bulk OA chemistry at MBO, the average HRMS of OA for
each of the three regimes are shown in Fig. S7. OA was generally highly oxidized under all three regimes and the
O/C of OA generally decreased as BB influence increased. In addition, ions larger than 100 amu ($f_{m/z>100}$)
contributed a larger fraction of the total organic signal during "BB Plm" periods (11%) compared to "No BB"
periods (5%), consistent with BBOA containing a larger fraction of high molecular weight compounds (Ge et al.,
2012a; Lee et al., 2016). OA in "No BB" air masses had an average O/C of 0.84 (O/C$_{AA}$, i.e., O/C calculated with
Aiken-Ambient method, is 0.63) and H/C of 1.48 (H/C$_{AA}$ = 1.29), in agreement with previous HR-AMS
measurements of free tropospheric OA at mountaintop sites (e.g., Sun et al., 2009; Rinaldi et al., 2015). The average
O/C for "BB Infl" and "BB Plm" periods were 0.77 (O/C$_{AA}$ = 0.60) and 0.69 (O/C$_{AA}$ = 0.53), respectively,
substantially higher than previously reported O/C for fresh BB emissions. For example, laboratory experiments
reported O/C$_{AA}$ in the range of 0.15 – 0.60 for POA from BB, depending on fuel type, burning condition, and burn
mass (Heringa et al., 2011; Ortega et al., 2013). The high O/C observed for BB-influenced OA at MBO indicates
that they were likely a combination of primary and secondary components with the secondary portion having a
substantial contribution to the bulk OA.
Changes in OA chemical composition due to wildfires is further investigated using the f$_{44}$ vs. f$_{60}$ plot (Fig. 4). All
OA data showed a progression where lower f$_{60}$ values were associated with higher f$_{44}$, consistent with aging of
BBOA observed both in laboratory studies and from airborne measurements (e.g., Cubison et al., 2011; Ortega et al.,
2013; Jolleys et al., 2015). f$_{44}$ during "No BB" periods spanned the range of 0.13 – 0.25 (mean = 0.17), due to the
dominance of highly oxidized OA. "BB Plm." data fell within the region defined by the BBOA measured previously
(Cubison et al., 2011; Ortega et al., 2013) and overlapped particularly well with fire plumes sampled above the
North America continent during the 2008 NASA ARCTAS mission and aged BBOA from controlled chamber open
burning of biomass (Cubison et al., 2011). Ambient fire plumes tended to have higher f$_{44}$ and lower f$_{60}$ values than
the POA from burning of various fuels in chamber studies (Ortega et al., 2013), mainly due to atmospheric aging.
However, the mixing of transported BB smoke with more oxidized background aerosols likely also contributed to
the changes in f$_{44}$ and f$_{60}$ observed for ambient BBOA. Furthermore, combustion conditions might also play a role in
how plumes map to the f$_{44}$ ~ f$_{60}$ space, as it has been shown in both ambient and chamber laboratory studies that
flaming-dominated fires for certain fuel types can lead to higher f$_{44}$ and are associated with lower f$_{60}$ compared to
more smouldering fires (Weimer et al., 2008; Jolleys et al., 2014; Collier et al., 2016).
**3.3   Aerosol source apportionment and contributions of primary and secondary BBOA at MBO**
To gain further insight into the influences of different sources and processes on OA concentration and
composition at MBO, we performed PMF analysis on the HRMS of all NR-PM$_1$ species acquired during this study.
PMF is commonly applied to the organic mass spectral matrix to determine distinct OA factors (Zhang et al., 2011
and references therein), but conducting PMF analysis on the combined spectra of organic and inorganic aerosols
allows for deriving additional information, e.g., the distributions of inorganic signals among different factors and the
nominal acidity of the factors, which benefits the interpretation of the sources, chemical characteristics, and
evolution processes of OA (Sun et al., 2012). For this study, a total of five OA factors were identified, including
three different BB-related aerosol types, i.e., BBOA-1 (O/C = 0.35), BBOA-2 (O/C = 0.60), and BBOA-3 (O/C =
1.06), and two distinct OOA factors, i.e., a less oxidized OOA associated with boundary layer (BL) dynamics
(BL-OOA, O/C = 0.69) and a more oxidized low-volatility OOA representing free-troposphere aerosol (LV-OOA,
O/C = 1.09). Unlike the two OOAs, the three distinct BBOA factors all showed high correlations with CO ($r^2$ = 0.70
– 0.86; Table S2) and displayed sporadic, high amplitude events with large enhancements in concentrations during
wildfire-influenced periods (Fig. 5a-c). In addition, the polar plots of all the BBOAs showed clear concentration
hotspots in the southwest direction at high wind speed (Fig. 6a-c), indicative of their associations with wildfire
plumes originating from SW Oregon and NW California (Fig. 1). Nevertheless, the three BBOAs are distinctly
different in terms of mass spectral profiles (Fig. 5k-m and Fig. S11), oxidation degrees, and volatility (Fig. 6g),
likely due to different extents of aging and/or processing pathways. Similarly, previous studies reported the
identification of multiple BBOA factors representative of different degree of atmospheric processing (e.g.,
Bougiatioti et al., 2014; Brito et al., 2014) and varying combustion conditions (e.g., Young et al., 2015; Young et al.,
2016). BBOA-1 and BBOA-2 looked more similar to the fresher BBOA factors while BBOA-3 was more similar to
the aged BBOA factors derived in Bougiatioti et al. (2014) and Brito et al. (2014) in terms of mass spectral features
(Fig, 4).
Among the three BBOA factors, BBOA-1 had the lowest O/C (0.35) and the highest H/C (1.76) and $f_{60}$ (2.2%)
(Fig. 5k). In addition, the mass spectrum of BBOA-1 showed prominent signals of $C_2H_3^+$, $CHO^+$, $C_4H_7^+$, $C_4H_9^+$, and
$C_9H_7^+$, markers for chemically-reduced aerosols, and a high abundance of ions larger than 100 amu ($f_{m/z>100}$ = 25%;
Fig. 5k and 5k'). The UMR spectrum of BBOA-1 at $m/z > 180$ exhibited a "picket fence" fragmentation pattern
where groups of peaks have 14 amu separation, suggesting the occurrence of molecules with hydrocarbon moieties
containing different units of the $CH_2$ group. The time series of BBOA-1 correlated tightly with those of $C_2H_4O_2^+$ and
$C_4H_9^+$ ($r^2$ = 0.94 and 0.95, respectively; Table S2), tracers for primary emissions. Furthermore, BBOA-1 appeared to
have a strong point source SW of MBO and peaked in association with high wind speeds suggesting that it could be
associated with plumes experiencing shorter transport times relative to plumes from equidistant fire sources (Fig.
6a). Together, these observations suggest that BBOA-1 was primarily associated with fresher and less processed air
masses from BB sources. In addition, BBOA-1 was found to be semivolatile (Fig. 6g), which is consistent with
previous findings that a majority (50% - 80%) of the POA in BB emissions is semivolatile (May et al., 2013). The
semivolatile behavior of BBOA-1 also explains the high degree of correlation between BBOA-1 and nitrate ($r^2$ =
0.60; Fig. 5a and Table S2), a secondary species that is often found to correlate with semivolatile OOA (SV-OOA)
(Zhang et al., 2011). However, despite being a secondary component, nitrate displayed tight correlations with
primary smoke markers, i.e., $C_2H_4O_2^+$ and $C_3H_5O_2^+$, at MBO (Fig. S12). Therefore, it appears that fast processing
near the fire sources led to the rapid conversion of $NO_x$ to more oxidized compounds such as PAN and nitrate.
Based on these results, we infer that BBOA-1 represents fresher BB emissions and might be a surrogate for primary
BBOA. On average, BBOA-1 comprised 20% of total OA mass during this study (Fig. 6f), suggesting that fresh BB
emissions exerted a significant impact on regional air masses.
The more oxygenated BBOA-2 (O/C = 0.60; H/C = 1.72) accounted for an average 17% of the total OA mass
(Fig. 6f). Its mass spectrum displayed characteristics of aged BBOA with lower abundances of $C_2H_4O_2^+$ ($f_{60}$ =
1.1%), $C_xH_y^+$ ions (31%), and ions > 100 amu ($f_{m/z>100}$ = 17%) compared to BBOA-1 (Figs. 4l, 4l' and S9b).
BBOA-2 also showed a somewhat less volatile profile compared to BBOA-1, especially at TD temperature < 150$^o$C
(Fig. 6g). In addition, the temporal trend of BBOA-2 displayed tight correlations with tracers for carboxylic acids,
e.g., $CHO_2^+$ and $CO_2^+$ ($r^2$ of 0.91 and 0.79, respectively; Fig. 5b and Table S2) but lower correlations with nitrate,
$C_2H_4O_2^+$, and $C_4H_9^+$. These results suggest that BBOA-2 was more chemically processed and likely contained

secondary products. Indeed, the polar plot of BBOA-2 (Fig. 6b) displayed a more dispersed pattern of sources compared to BBOA-1 with hotspots located in various directions. Nevertheless, the occurrence of a high concentration band at 5 - 15 m s$^{-1}$ in the SW direction suggests important BBOA-2 sources from similar distances and locations as BBOA-1. The dispersed source features are further evidence that BBOA-2 is more secondary in nature compared to BBOA-1 and is likely more aged.

BBOA-3 contrasts strongly with BBOA-1 and BBOA-2 in chemical composition. The HRMS of BBOA-3 had a very low $C_2H_4O_2^+$ signal ($f_{60} = 4 \times 10^{-8}$), a relatively high intensity of $CO_2^+$ ($f_{44} = 0.215$) and a high degree of oxidation (O/C = 1.07; Fig. 5m), all of which highly resemble those of LV-OOA (Fig. 5o). However, the mass spectra at large $m/z$'s indicated distinct chemical differences between BBOA-3 and LV-OOA (Fig. 5m' and 5o'), as there appeared to be a higher abundance of high molecular weight species in BBOA-3. In addition, the temporal variation patterns of BBOA-3 and LV-OOA were dramatically different ($r^2 = 0.07$) and BBOA-3 closely correlated with CO ($r^2 = 0.86$; Fig. 5c and Table S2) whereas LV-OOA did not ($r^2 = 0.008$). As shown in Fig. 6, the polar plot of BBOA-3 showed a high concentration band from SW at a wind speed of $5 – 15$ m s$^{-1}$, which overlaps with the hot spot shown in the BBOA-1 polar plot (Fig. 6a). These results indicate that BBOA-3 was associated with wildfires and likely formed both through rapid processing near the wildfire source and during transport to MBO. However, given that humic-like substances (HULIS) are a known component of BB emissions and that these substances resemble BBOA-3 in terms of AMS mass spectrum, high degree of oxygenation, and low volatility (Dinar et al., 2006; Adler et al., 2011), it is possible that a fraction of BBOA-3 was HULIS as well.

Another important characteristic of BBOA-3 is that it appeared to be composed of some very low-volatility compounds. As shown in Fig. 6g, ~ 60% of its mass remained in the aerosol phase at a temperature of 200 °C. This observation is consistent with previous studies which have observed the presence of low-volatility and extremely low volatility BBOA materials in aged wildfire plumes (Lee et al., 2016; Paciga et al., 2016) and in SOA produced from major organic gases from BB (e.g., phenols) (Yu et al., 2016). It is important to note that the highly oxidized BBOA-3 on average accounted for 31% of the total OA mass during this study, which implies that a significant fraction of the highly aged BBOA may appear indistinguishable from OOA from other sources due to mass spectral similarities (e.g., low $f_{60}$ and high $f_{44}$) and hence would lead to an underestimation of the influence of BB emissions on a regional scale.

BL-OOA and LV-OOA accounted for the remaining 32% of total OA mass during this study. These two OOAs were not associated with BB, as indicated by low $f_{60}$ (Fig. 5n and 5o) and a lack of correlation with BB tracers (Table S2). BL-OOA was relatively oxidized (O/C = 0.69; Fig. 5n) and appeared significantly less volatile than nitrate but more volatile than sulfate (Fig. 6g). BL-OOA showed a distinct diurnal cycle highly resembling that of water vapor (Fig. 5i), which is a tracer for BL upslope flow during the daytime at MBO (Weiss-Penzias et al., 2006). Photochemical production of OA in the early afternoon may also contribute to the daytime increase of BL-OOA. Furthermore, the time series of BL-OOA correlated with $CH_3SO_2^+$ (Fig. 5d and Table S2), a signature ion for methanesulfonic acid (MSA) (Ge et al., 2012b). MSA is typically associated with marine sources but has been found to have terrestrial sources as well (Ge et al., 2012b; Young et al., 2016). All these results suggest the influence of BL dynamics on BL-OOA. In comparison, the LV-OOA was highly oxidized (O/C = 1.09) with a pronounced $CO_2^+$

peak in the spectrum (Fig. 5o). In addition, Fig. 6g indicates that LV-OOA shared a similar volatility profile as
sulfate, showing no sign of evaporation until the TD temperature reached nearly 130°C, consistent with LV-OOA
previously determined in other ambient studies (Huffman et al., 2009; Paciga et al., 2016). The diurnal pattern of
LV-OOA appeared to be rather flat (Fig. 5j) and its polar plot had the most dispersed feature among all factors (Fig.
6e). All these observations suggest that this factor is representative of free tropospheric aerosol.

### 3.4  A case study of the aging of BBOA in wildfire plumes

Based on MODIS fire hotspot information, the Salmon River Complex fire (SRCF) was continuously burning
from August 13 to August 17 (Fig. 7a). Three-day HYSPLIT back trajectories suggest that air masses arriving at
MBO from August 14 22:00 to August 16 09:00 passed over the SRCF (Fig. 7a), consistent with the observations of
persistent SW wind at MBO during this period (Fig. 7c). MODIS also detected a few hotspots from the Whiskey
Complex Fire (~ 43°N, 122.8°W) intermittently on August 15 but the fire was much weaker compared to SRCF as
indicated by the lower fire radiative power (FRP, Fig. 7a). We therefore assume that the emissions arriving at MBO
during this time period were from a single source and therefore consistent in transport distance and fuel type.
Combining MODIS fire hotspots and back-trajectories, we estimated that the transport time of SRCF plumes ranged
from 8 to 11 hours before being sampled at MBO.
In order to examine how atmospheric aging affects BBOA chemistry, we calculated cumulative solar radiation
($\sum$SR) and average RH over the total transport time (from source to MBO) for each trajectory and plotted them
versus air mass arrival time in Fig. 7b. $\sum$SR denotes the total amount of solar radiation that the smoke plumes were
exposed to during transport and can be used as an indicator for the extent of photochemical aging assuming the
plumes were optically thin. RH in the air mass history was relatively stable, however $\sum$SR clearly varied throughout
the measurement period such that some BB plumes experienced more solar radiation than others and some were
transported exclusively at night. Furthermore, the burn conditions were modestly constant during this period with an
average modified combustion efficiency (MCE) value of 0.88 (± 0.03) for the BB plumes that met the criteria for
MCE calculation (Collier et al., 2016). Furthermore, the MCE values showed no differences between nighttime and
daytime plumes and didn't correlate with $\sum$SR (Fig. S13). These conditions, together with the high emissions
concentrations for both gas and particle phase components (Fig. 7d - f), provide a near ideal case study where
atmospheric aging is likely the largest factor affecting the chemical evolution of BBOA.
During this SRCF case study period, CO, $NO_y$, and PAN mixing ratios observed at MBO exhibited similar trends
that varied dynamically and correlated well with the fresh BBOA-1 factor (Fig. 7d - f). In addition, OA was
overwhelming dominated by BBOAs, which summed to contribute 80% - 99% of total OA mass (Fig. 7g). The
chemical parameters of OA and the fractional contributions of each BBOA factor appear to be related to $\sum$SR (Figs.
6g and 6h). In order to investigate the chemical evolution of BBOA, we reconstructed the time series and the
chemistry parameters of total BBOA (= BBOA-1 + BBOA-2 + BBOA-3) from the residual matrix of organic
aerosol after subtracting the contributions from BL-OOA and LV-OOA. The carbon oxidation state ($OS_c = 2 \times$ O/C
– H/C; (Kroll et al., 2011)) of total BBOA showed a clear increasing trend with respect to $\sum$SR, consistent with the
trends of O/C and $f_{44}$, while H/C, $f_{60}$, and $f_{m/z>100}$ of total BBOA showed decreasing trends with $\sum$SR (Fig. 8). The
relationship between $f_{44}$ and $f_{60}$ for total OA observed during this case study is shown in Fig. 4. $f_{60}$ decreased with
increased $f_{44}$ due to aging and the data overlapped with the aged BBOA from controlled chamber open burning of
turkey oak (Cubison et al., 2011). These results suggest oxidation of anhydrous sugar and other BBOA components
due to photochemical aging, consistent with previous observations in the laboratory (Grieshop et al., 2009;
Hennigan et al., 2011; Ortega et al., 2013) and field (Cubison et al., 2011; May et al., 2015). In addition, the
negative correlation between BBOA-1 and $\sum SR$ and the positive correlations of BBOA-2 and BBOA-3 with $\sum SR$
(Fig. 8) corroborated our earlier assumption that BBOA-2 and BBOA-3 represented more aged, secondary BBOA
whereas BBOA-1 represented primary BBOA.
We classify the plumes according to $\sum SR$ and designate those as night-time transported if $\sum SR$ was below 500 W
$m^{-2}$, and the rest as day-time transported. OA concentration and CO mixing ratio were tightly correlated, with $r^2$ =
0.88 and 0.94 for night- and day-time transported plumes, respectively (Fig. 9a). CO has been commonly used as a
stable plume tracer to account for dilution and the slope obtained from orthogonal fitting between OA and CO is
defined as the enhancement ratio (i.e., $\Delta OA/\Delta CO$). Change of $\Delta OA/\Delta CO$ during plume transport indicates the
influence of factors other than dilution, e.g., SOA formation or OA evaporation. For the SRCF case study,
$\Delta OA/\Delta CO$ was very similar for the day-plumes and the night-plumes: $0.28 \pm 0.014$ vs. $0.27 \pm 0.005$ $\mu g\ m^{-3}\ ppbv^{-1}$
respectively (Fig. 9a), suggesting no net OA mass enhancement due to photochemical aging. This is consistent with
the findings of Collier et al. (2016), which compared selected BB events from this dataset measured at MBO to
those aboard a research aircraft sampling fresher plume emissions and found very similar OA enhancements
between the fresher and more aged emissions. However, compared to daytime-plumes, OA for plumes transported
during night time was less oxidized (Fig. 9c and 9d) and was dominated by the fresh BBOA-1 (53%), followed by
the most oxidized BBOA-3 (24%), and intermediately oxidized BBOA-2 (15%; Fig. 9b). By contrast, daytime
plumes were characterized by a significant decrease in the mass fraction of BBOA-1 (37%) coupled with increases
in the BBOA-2 (20%) and BBOA-3 (37%). This is corroborated by the significant differences in chemical
composition for the two types of plumes, where the average HRMS (Fig. 9c and 9d) indicated that the BBOA in
day-time plumes had a higher degree of oxidation (average O/C = 0.66) compared to the night plumes (O/C = 0.55).
These observations together suggest that although net OA production was conserved with higher photochemical
aging, BBOA was chemically transformed, likely due to oxidative processing in both gas and particles phases
followed by fragmentation and volatilization.
**4    Summary and conclusions**
We have characterized the chemical composition and properties of aerosols at a high elevation site that was
heavily impacted by wildfire smoke plumes in the western US during the BBOP campaign in summer 2013. The
sampling site was located on the summit of Mt. Bachelor, an isolated volcanic peak, in central Oregon. It was
impacted by regional wildfire emission during a majority of the campaign and saw intense BB plumes with elevated
air pollutants (up to 700 ppbv of CO and $\sim 210\ \mu g\ m^{-3}$ of NR-PM$_1$). The average ($\pm 1\sigma$) NR-PM$_1$ mass concentration
was 22.4 ($\pm 17.7$) $\mu g\ m^{-3}$ during fire-impacted periods, mostly due to OA that dominated the NR-PM$_1$ composition.
In contrast, the average NR-PM$_1$ concentration was only 3.7 $\mu g\ m^{-3}$ over periods free of BB influence and the
aerosols contained a high mass fraction of ammonium sulfate (up to ~ 90%). In addition to increasing regional
aerosol concentrations, wildfires in the Pacific Northwest region also significantly increased the mixing ratios of CO,
$NO_y$, and PAN, although $NO_x$ and $O_3$ displayed more complex behavior.
PMF analysis identified three types of BBOA that together accounted for 68% of the OA mass during this study,
in addition to two types of OOA representing regional air conditions. The time series of all BBOA factors displayed
dynamic variations that tightly correlated with those of CO and aerosol light scattering. Yet the three BBOAs were
significantly different both chemically and physically and appeared to have been subjected to different degrees of
atmospheric processing. BBOA-1 appeared to represent fresh wildfire emissions and featured semivolatile behavior,
low O/C, a larger fraction of anhydrous sugar ($f_{60}$ = 2.2%), and a strong association with active wildfire sources. On
the other hand, BBOA-2 and BBOA-3 represented more aged BB emissions and showed higher oxidation degree,
lower $f_{60}$, significantly lower volatilities, and more dispersed source regions. BBOA-3, in particular, had an O/C of
1.06, very low volatility, and almost no contribution from $f_{60}$, and thus appeared to be chemically similar to highly
oxidized SOA observed in the atmosphere. Nevertheless, BBOA-3 is substantially different than the LV-OOA factor
identified in this study; in addition to dramatically different temporal variation patterns, BBOA-3 also seemed to be
composed of a higher fraction of high molecular weight species as well as compounds of extremely low volatilities.
A case study using consecutive BB plumes transported from the same fire source was performed to examine in
detail the environmental factors leading to BBOA evolution. The BB plumes were associated with fires of similar
modified combustion efficiencies but were exposed to a wide range of photochemical aging, as indicated by the
cumulative solar radiation along the trajectory history from fire source to the sampling site. The results showed that
photochemical aging led to more oxidized OA with higher mass fractions of aged BBOA (i.e., BBOA-2 and BBOA-
3) and a lower fraction of fresh BBOA-1. Although BBOA in daytime plumes were chemically more processed than
nighttime plumes, the enhancement ratios of OA relative to CO were very similar under the night-time and day-time
conditions ($\Delta OA/\Delta CO$ = 0.28 ± 0.014 and 0.27 ± 0.005 $\mu g\ m^{-3}\ ppbv^{-1}$, respectively). One explanation for this
apparent lack of net SOA production in transported BB plumes is that SOA formation in BB emissions was balanced
by POA loss, likely due to oxidation followed by fragmentation and volatilization.
Over the entire period of this study, the aged BBOA-2 and BBOA-3, most of which were likely secondary, on
average made up ~ 50% of the OA mass observed at MBO. Aged BBOAs were present at significant concentrations
even in relatively fresh plumes (~ 6 - 12 hr of atmospheric aging). These results suggest that BB emissions undergo
substantial chemical processing which commences directly after emission and continues during atmospheric
transport, forming and transforming aerosols that can significantly influence air quality and atmospheric chemical
composition at downwind sites with important implications for health and climate.
**Acknowledgements**
This work was funded by US Department of Energy (DOE) Atmospheric System Research (ASR) Program (DE-
SC0014620 and DE-SC0007178). Shan Zhou was partially funded by a PhD grant from the Chinese Scholarship
Council (CSC) and the Donald G. Crosby Fellowship at UC Davis. We acknowledge the use of MODIS fire hotspot
data and imagery from LANCE FIRMS, downloadable from https://firms.modaps.eosdis.nasa.gov and operated by

the NASA/GSFC/Earth Science Data and Information System (ESDIS) with funding provided by NASA/HQ. Special acknowledgement goes to Mt. Bachelor Summit Ski Lift technicians, Advanced Northwest Welding, LLC, and our lab members, Caroline Parworth, Xinlei Ge, and Jianzhong Xu, whose help was invaluable in setting up logistics for site sampling. The MBO is supported by a grant to the University of Washington from NSF (NSF# 1447832).

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

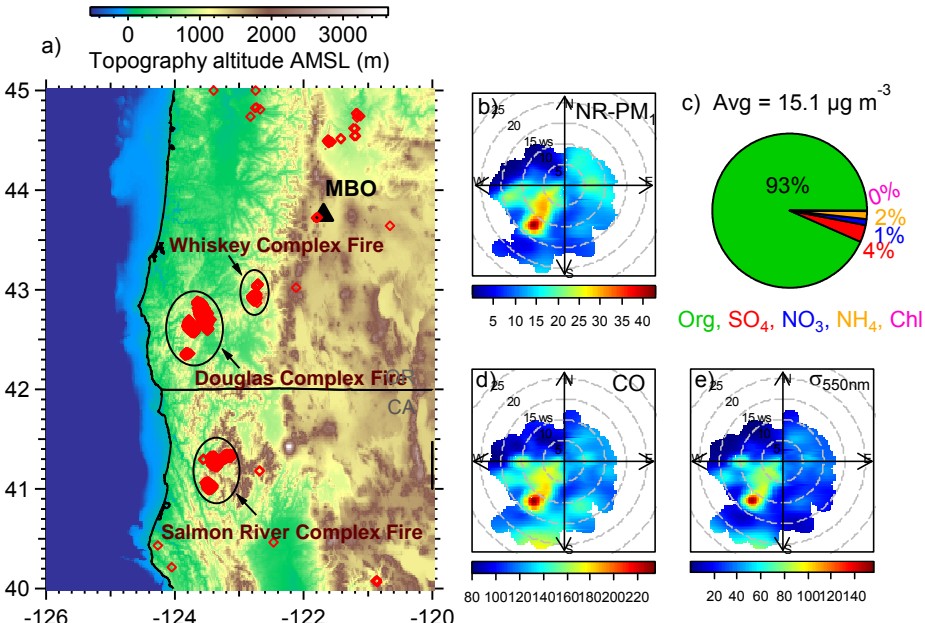


**Fig. 1.** a) Map with MBO (black solid triangle) and wildfires detected by MODIS (red open diamonds) in the
Northwest Pacific US from July 25 to August 25, 2013. Three fire complexes, Whiskey Complex Fire (WCF),
Douglas Complex Fire (DCF), and Salmon River Complex Fire (SRCF) are highlighted with black circles. Bivariate
polar plots of (b) NR-PM$_1$ concentrations (in µg m$^{-3}$), d) submicrometer aerosol light scattering at 550 nm ($\sigma_{550nm}$ in
Mm$^{-1}$) and (e) CO mixing ratio (in ppbv) during the sampling period. c) Average NR-PM$_1$ composition for the
sampling period.

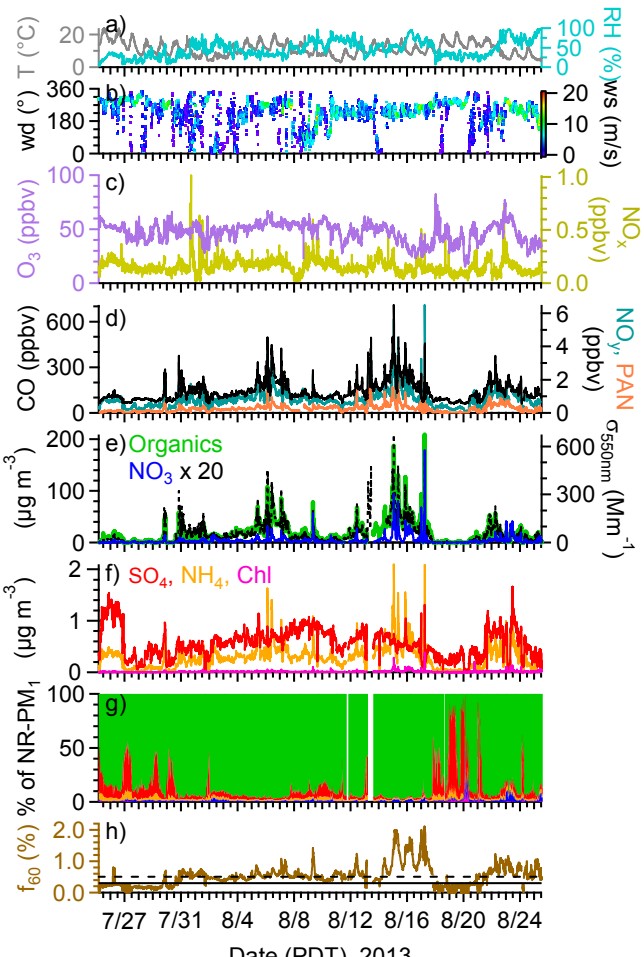


**Fig. 2.** Time series of (a) temperature (T) and relative humidity (RH), (b) wind direction (WD) colored by wind
speed (WS), (c) mixing ratios of $O_3$ and $NO_x$, (d) mixing ratios of CO, $NO_y$ and PAN, (e and f) mass concentrations
of NR-PM$_1$ species and $\sigma_{550nm}$ in STP (T = 273 K, P = 1013.25 hPa), (g) NR-PM$_1$ composition, and (h) $f_{60}$ (=
$C_2H_4O_2^+$ / OA). The solid and broken lines in (h) indicate $f_{60}$ = 0.3% and $f_{60}$ = 0.5%, respectively.

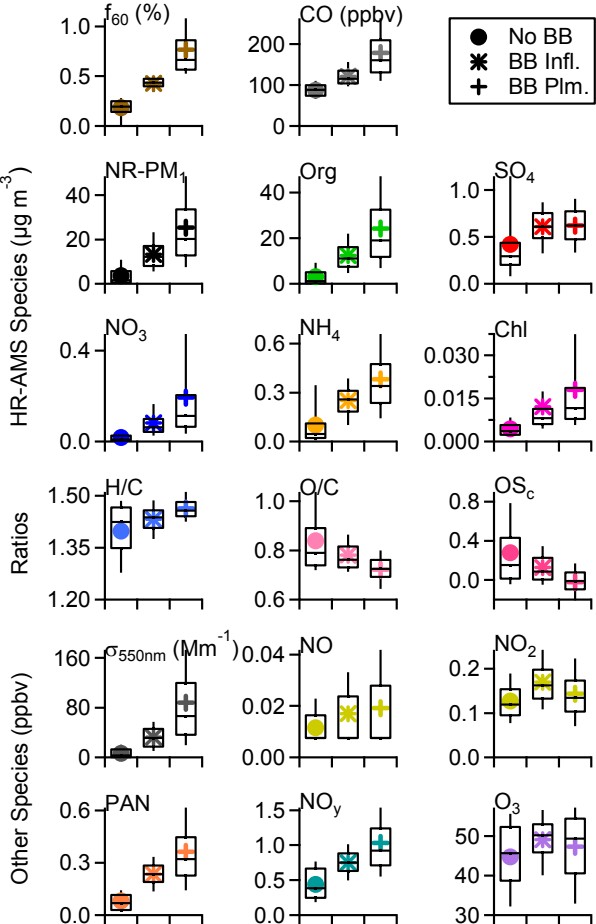


**Fig. 3.** Box plots that compare $f_{60}$ values, CO mixing ratios, NR-PM$_1$ species concentrations, OA elemental ratios,
carbon oxidation states (OS$_c$), $\sigma_{550nm}$, and mixing ratios of trace gases among three aerosol regimes ("No BB", "BB
Infl", and "BB Plm"). The whiskers indicate the 90th and 10th percentiles, the upper and lower boundaries indicate
the 75$^{th}$ and 25$^{th}$ percentiles, and the lines in the boxes indicate the median values and the markers indicate the mean
values.

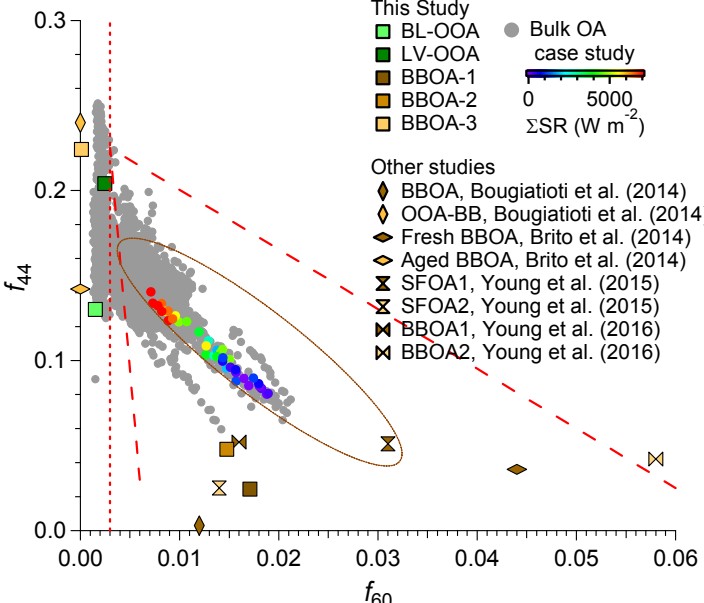


**Fig. 4**. Scatter plot of $f_{44}$ vs. $f_{60}$. The grey markers correspond to the measured OA during this study and the Salmon River Complex Fire (SRCF) case study data are colored by cumulative solar radiation ($\Sigma$ SR). In addition, the five OA factors identified in this study are shown as solid squares and the BBOA factors reported in literature where multiple BBOA factors were derived are shown in different markers. The dashed red lines denote $f_{60}$ = 0.003 and the boundaries set for BBOA (Ortega et al., 2013). The brown oval encompasses ARCTAS fire plumes sampled above the North America Continent (Cubison et al., 2011).

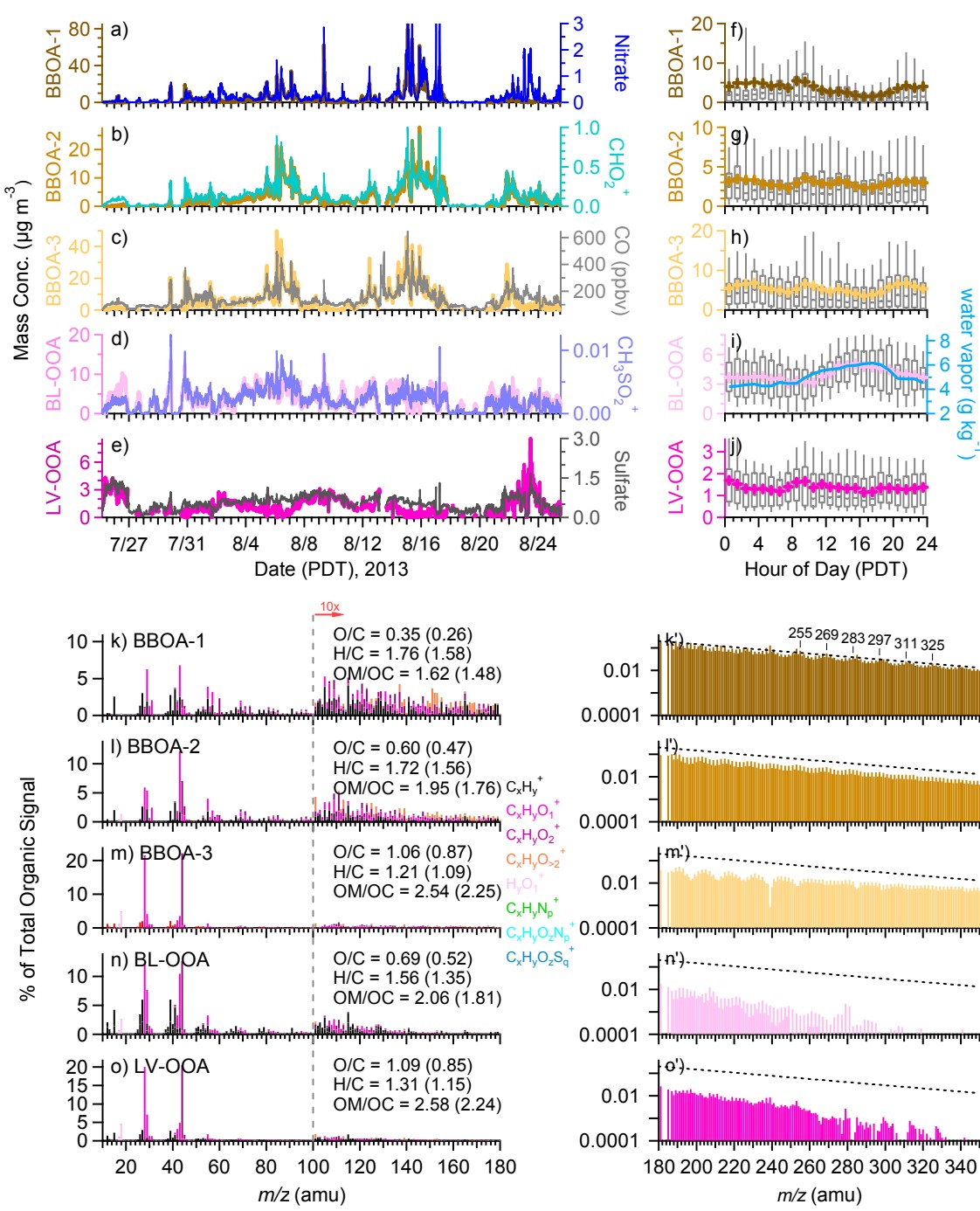

**Fig. 5.** (a-e) Time series of OA factors and corresponding tracer compounds. Organic ions are in organic equivalent mass; (f-g) Diurnal variations of OA factors (the whiskers above and below the boxes indicate the 90th and 10th percentiles, the upper and lower boundaries indicate the 75th and 25th percentiles, and the lines in the boxes indicate the median values and the cross symbols indicate the mean values) with the diurnal cycle of mean water vapor in (i); (k-o) HRMS of OA factors colored by eight ion families at $m/z$ < 180 and (k'-o') UMR MS at $m/z$ > 180 for each OA factor. The elemental ratios of each OA factor are shown in the legends of (k-o) with those obtained using the AA method in parenthesis.

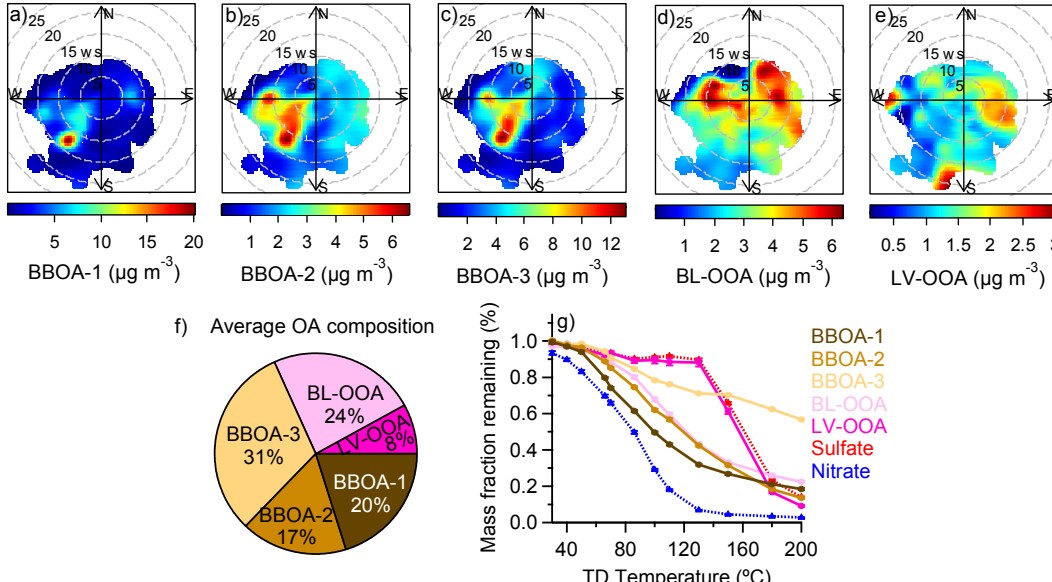


**Fig. 6.** (a-e) Bivariate polar plots that illustrate the variations of the concentrations of each OA factor as a
function of wind speed (m s$^{-1}$) and wind direction; (f) Average OA composition during the sampling period; (g)
Volatility profiles of OA factors, sulfate, and nitrate, with error bars showing the standard deviation of the
calculated slope, i.e., mass fraction remaining.

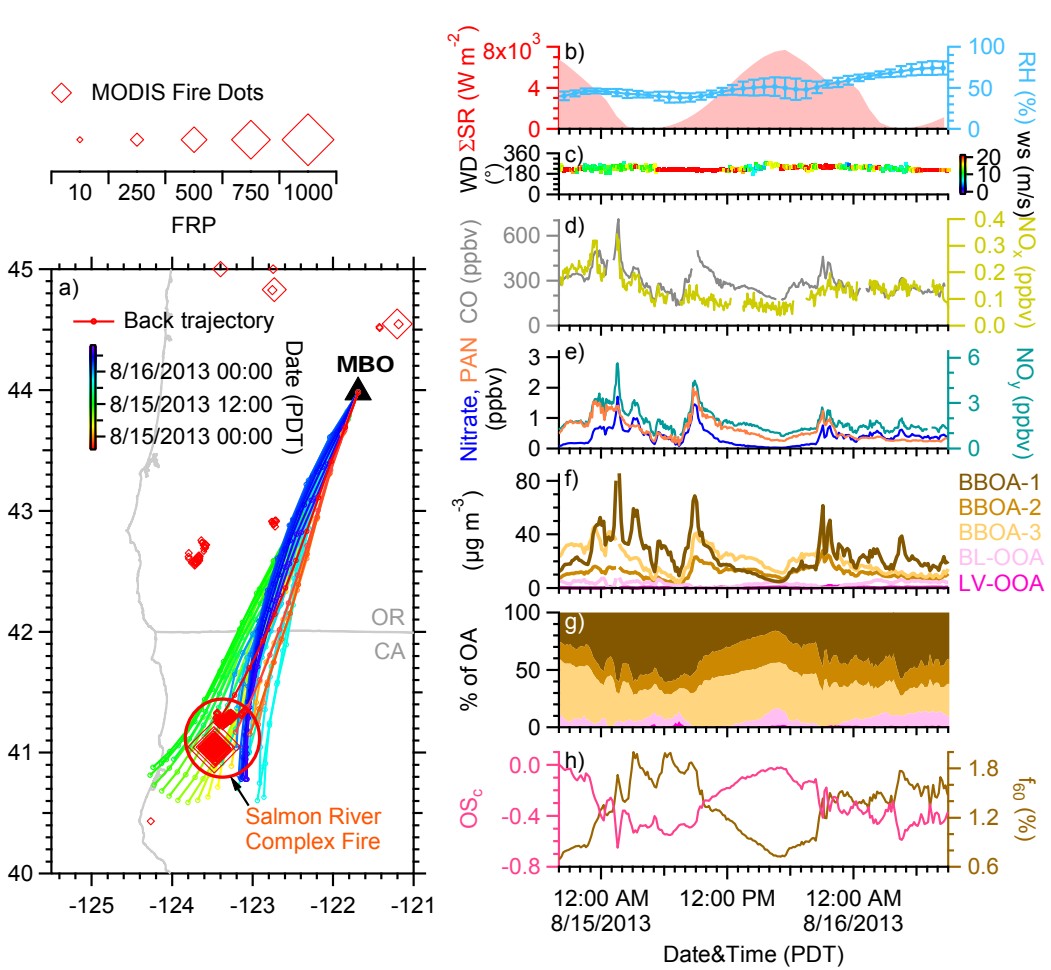


**Fig. 7.** (a) Map of the Pacific Northwest with the location of MBO marked by black triangle. Open diamonds represent MODIS satellite fire dots detected during August 13 – 17, 2013, and are sized by fire radiative power (FRP). Twelve-hour HYSPLIT back trajectories of air masses arriving at MBO from August 14 20:00 to August 16 09:00 are colored by time of arrival at MBO. Markers indicate 1-hour interval; (b) Cumulative solar radiation ($\sum$SR) and average RH for each trajectory; (c) Wind direction (WD) colored by wind speed (WS) measured at MBO; Mixing ratios of (d) CO, $NO_x$, (e) nitrate, PAN, and $NO_y$; (f) Five OA factors; (g) OA composition; (h) Average carbon oxidation states and $f_{60}$ of OA during the Salmon River Complex Fire (SRCF) case study period.

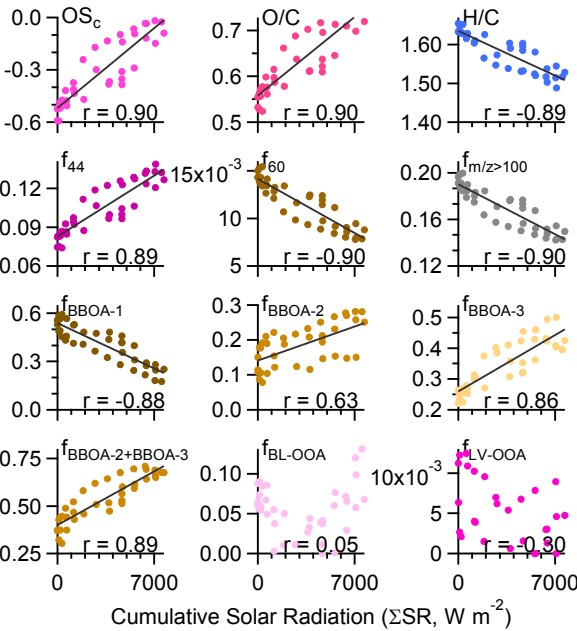


**Fig. 8.** Aerosol chemistry parameters of total BBOA as a function of cumulative solar radiation for the Salmon

River Complex Fire case study. The Pearson's correlation coefficients (r) are reported.

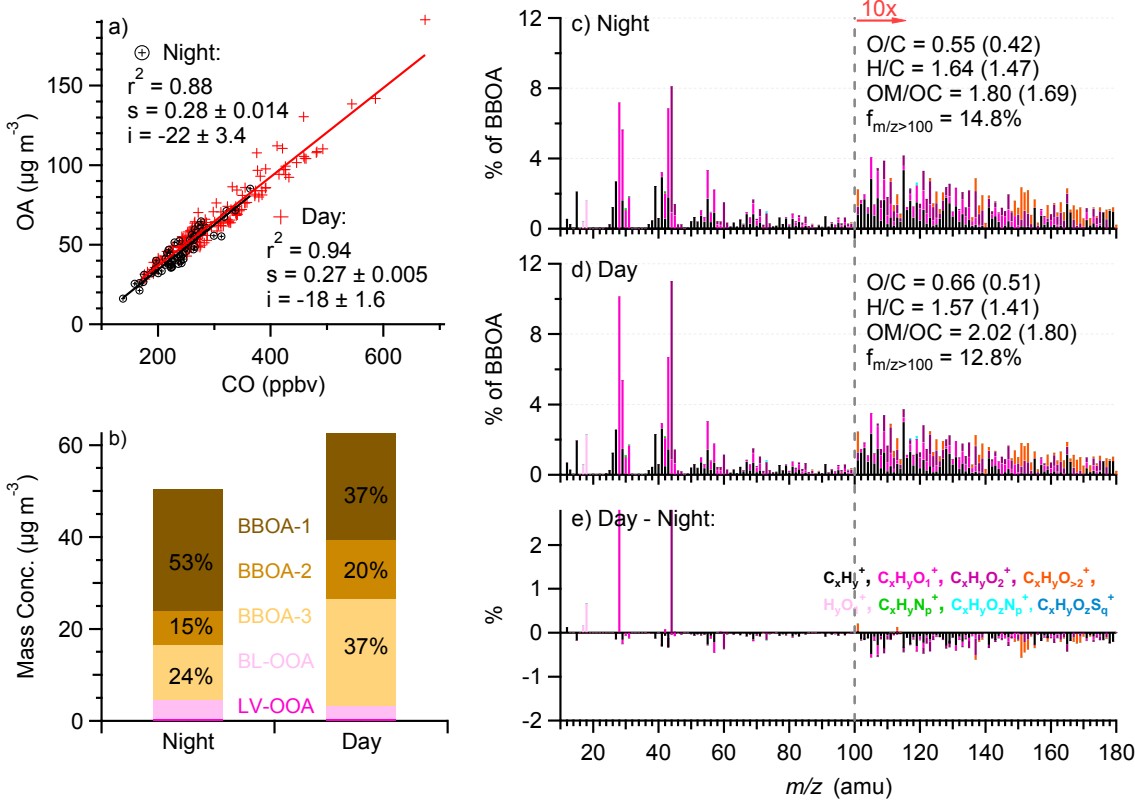


**Fig. 9.** (a) OA vs. CO during August 14 20:00 to August 16 09:00 with night-time transported plumes illustrated
as black circles and day-time transported plumes as red crosses. The orthogonal distance regression (ODR) results
for the two plume types are shown with the 1-σ uncertainties reported for the fit slopes (s) and intercepts (i); (b) A
comparison of the average concentrations of 5 OA factors (stacked) between the night-time and day-time transported
plumes. The average mass fractions of the BBOAs to total OA mass in each plume type are reported; (c) Average
HRMS of total BBOA for the night-time transported plumes; (d) Average HRMS of total BBOA for the day-time
transported plumes and (e) Difference BBOA HRMS between day and night plumes. The elemental ratios of BBOA
calculated with the IA method are shown in the legends of (c) and (d) with those obtained using the AA method in
parenthesis.