# Peer review of "Regional Influence of Wildfires on Aerosol Chemistry in the 1"

_Atmospheric Chemistry and Physics, 2016_

## Referee Comment (RC1) · Anonymous Referee #1 · 11 Oct 2016

This is a paper concerning AMS observations of biomass burning smoke during BBOP from a gound-based site. Detailed observations are systematically reported and factorisation is performed, yielding three factors related to biomass burning. The factorisation and an analysis of volatility offer some new and interesting insights and there is a case study looking at trends with atmospheric ageing time, which sheds new light on the aging timescales of these aerosols.

While this isn't the first paper on this general topic, the depth of analysis does provide some new insights, so it is within the scope of ACP. It is also very well written in general and comes with a decent amount of supplementary information supporting the methods used in factorisation (something that is lamentably absent from many papers). I

recommend this be published subject to the following comments:

General comments

I generally found that comparisons with previous works in the literature to be inadequate. While there are certainly novel aspects of this work, there have been a number of papers published previously on this topic and yet most of these are only given a cursory mention to say that the results here are qualitatively consistent. As cited in the manuscript, there have been a number of papers reporting the ageing of biomass burning with an AMS (e.g. Cubison et al., 2011), so a more quantitative comparison should be possible here. Also, there have been other studies reported where multiple BBOA factors have been derived (e.g. http://www.atmos-chem-phys.net/15/2429/2015/), so a detailed comparison should be possible there. By placing the results here in the wider context better, this will improve the quality of the paper's conclusions.

Specific comments

Line 152: The 1.5 $\mu$g/m3 cutoff is not adequately justified and seeing as many AMS factorisations have been performed successfully with signals lower than this, it would appear to be an odd thing to do. Why was this chosen as the cutoff? What were the actual S/Ns, according to the error model? Why wasn't the low S/N data simply downweighted, as is standard practice? It seems to me that the real benefit of this strategy is so that the factorisation will be of the high-intensity plumes (the subject of interest) than background data (which can hinder convergence not through low S/N but through rotational ambiguity). This is, in its own way, justifiable, but it should be presented as such, rather than a simple S/N issue.

Line 207: How was potassium measured? This is important because it affects the credibility of the data. What constitutes 'low'? A quantitative comparison with other studies should be given.

Line 276: While a combined approach to PMF can improve factorisation and aid interpretation, it can also harm the analysis. The fact that the sources and processes governing inorganics are often fundamentally different to those of organics means that their inclusion can introduce 'model error', in turn increasing rotational ambiguity. Also, because of the high 'strength' of the inorganic variables (owing to their relative lack of fragmentation), they may lead the factors, causing the organic data to more reflect inorganic, rather than organic, sources and processes. This note of caution should be added to the text. Did the authors try running PMF without the inorganics? How did the solutions differ?

Line 330: The results here do not necessarily prove that BBOA-3 was formed through processing. While the addition of oxygen would be consistent with other observations of ageing in biomass burning, it does not discount the possibility that an amount of it was in the form of primary humic-like substances, which are known to be formed during biomass burning (e.g. http://www.atmos-chem-phys.net/6/5213/2006/). These generally bear a resemblance to highly oxygenated secondary organic aerosol in terms of their volatility, chemical functionality and AMS mass spectra, so a caveat should be added.

---

## Referee Comment (RC2) · Anonymous Referee #2 · 16 Nov 2016

General comments:

This manuscript reports HR-AMS measurements of fresh and aged biomass burning emissions observed from the Biomass Burning Observation Project (BBOP) field campaign in summer 2013. PMF analysis and other measurements were performed to investigate atmospheric chemistry of biomass burning organic aerosol (BBOA) in Western US. This study observed that all BBOA factors (BBOA-1, 2 and 3) composed of a larger fraction of high molecular weight organics compared to oxygenated organic aerosols factors (BL-OOA and LV-OOA). Thermodeunder measurements further suggested the presence of low-volatility BBOA in aged biomass burning plume, which is consistent to some recent literature. More importantly, a case study provides insight

into the net production of organic aerosol mass due to atmospheric aging of wildfire plume, which is of great interest to the atmospheric aerosol community. The measurements and data analysis were well performed and the major scientific arguments are convincing. The manuscript is well organized and written in general. I recommend this manuscript to be published in Atmospheric Chemistry and Physics after addressing the specific comments below.

Specific comments:

1. PMF analysis, Line 149-156: It is uncommon to run PMF with inorganic components as those peaks could be too strong that drive the overall PMF solution. Therefore, it is recommended to better highlight the merits and rationales behind to include inorganic components in the PMF analysis, and briefly compare their existing PMF results to that without inorganic fragments.

2. Potassium detection, line 207-211: The author mentioned that the potassium signal was low throughout the whole period of study but it is more important to examine if the temporal variation of potassium correlates with those of the identified BBOA factors. In addition, potassium background in AMS data is high in general due to surface ionization of tungsten vaporizer. Please report detection limit of potassium and compared to the ambient data.

3. Ammonium level in the BB plume, line 235: Figure S7 shows that the concentrations of ammonium were much higher than that required to completely neutralize sulfate, nitrate and chloride when organic loadings were high (e.g. > 50 ug/m3) due to the presence of biomass burning plume. It is well-known that biomass burning can produce significant amounts of nitrogen-containing organics such as amine. Please discuss if the observed NHx fragments in biomass burning plume were due to the increased level of amine in particle phase.

Minor and technical comments:

[Figure]

1. Line 58-60: Please add the recent publication by Gilardoni et al. (2016) that reports SOA formation in the aged biomass burning emission through aqueous-phase processing.

2. Line 158: It is unclear whether time-dependent or average CE applied to the PMF results.

3. Line 167-169: Uncertainties of the mass fraction remaining (MFR) for each factor are increasing with the operating temperature of thermodenuder, especially for more volatile species. Please highlight the potential uncertainties in the revised version.

4. Line 345: It should be "Fig. 4i" instead of "Fig. 5i".

References:

Gilardoni Stefania, Massoli Paola, Paglione Marco, Giulianelli Lara, Carbone Claudio, Rinaldi Matteo, Decesari Stefano, Sandrini Silvia, Costabile Francesca, Gobbi Gian Paolo, Pietrogrande Maria Chiara, Visentin Marco, Scotto Fabiana, Fuzzi Sandro, and Facchini Maria Cristina (2016). Direct observation of aqueous secondary organic aerosol from biomass-burning emissions. Proceedings of the National Academy of Sciences of the United States of America 113: 10013-10018.

---

## Author Comment (AC1) · 2 Jan 2017

**Response to Reviewers for Paper acp-2016-823**

**Regional Influence of Wildfires on Aerosol Chemistry in the Western US and Insights into Atmospheric Aging of Biomass Burning Organic Aerosol**
By Shan Zhou et al.

We thank the reviewers for their thoughtful comments. We have carefully revised the manuscript accordingly. Our point-to-point responses can be found below, with reviewer comments repeated in black and author responses in blue. Changes made to the manuscript are in quotation marks.

**Author Responses to Anonymous Referee #1**

This is a paper concerning AMS observations of biomass burning smoke during BBOP from a gound-based site. Detailed observations are systematically reported and factorisation is performed, yielding three factors related to biomass burning. The factorisation and an analysis of volatility offer some new and interesting insights and there is a case study looking at trends with atmospheric ageing time, which sheds new light on the aging timescales of these aerosols.

While this isn't the first paper on this general topic, the depth of analysis does provide some new insights, so it is within the scope of ACP. It is also very well written in general and comes with a decent amount of supplementary information supporting the methods used in factorisation (something that is lamentably absent from many papers). I recommend this be published subject to the following comments:

General comments

I generally found that comparisons with previous works in the literature to be inadequate. While there are certainly novel aspects of this work, there have been a number of papers published previously on this topic and yet most of these are only given a cursory mention to say that the results here are qualitatively consistent. As cited in the manuscript, there have been a number of papers reporting the ageing of biomass burning with an AMS (e.g. Cubison et al., 2011), so a more quantitative comparison should be possible here. Also, there have been other studies reported where multiple BBOA factors have been derived (e.g. http://www.atmos-chem-phys.net/15/2429/2015/), so a detailed comparison should be possible there. By placing the results here in the wider context better, this will improve the quality of the paper's conclusions.

In response to this comment, we have added a new figure (Fig. 4) and related discussions in various places in the revised manuscript:

In the last paragraph in section 3.2: "Changes in OA chemical composition due to wildfires is further investigated using the $f_{44}$ vs. $f_{60}$ plot (Fig. 4). All OA data showed a progression where lower $f_{60}$ values were associated with higher $f_{44}$, consistent with aging of BBOA observed both in laboratory studies and from airborne measurements (e.g., Cubison et al., 2011; Ortega et al., 2013; Jolleys et al., 2015). $f_{44}$ during "No BB" periods spanned the range of 0.13 - 0.25 (mean = 0.17), due to the dominance of highly oxidized OA. "BB Plm." data fell within the region defined by the BBOA measured previously (Cubison et al., 2011; Ortega et al., 2013) and overlapped particularly well with fire plumes sampled above the North America continent during the 2008 NASA ARCTAS mission and aged BBOA from controlled chamber open burning of biomass (Cubison et al., 2011). Ambient fire plumes tended to have higher $f_{44}$ and lower $f_{60}$

values than the POA from burning of various fuels in chamber studies (Ortega et al., 2013), mainly due to atmospheric aging. However, the mixing of transported BB smoke with more oxidized background aerosols likely also contributed to the changes in $f_{44}$ and $f_{60}$ observed for ambient BBOA. Furthermore, combustion conditions might also play a role in how plumes map to the $f_{44} \sim f_{60}$ space, as it has been shown in both ambient and chamber laboratory studies that flaming-dominated fires for certain fuel types can lead to higher $f_{44}$ and are associated with lower $f_{60}$ compared to more smouldering fires (Weimer et al., 2008; Jolleys et al., 2014; Collier et al., 2016).

In the third paragraph in section 3.4: "…The relationship between $f_{44}$ and $f_{60}$ for OA observed during this case study is shown in Fig. 4. $f_{60}$ decreased with increased $f_{44}$ due to aging and the data overlapped with the aged BBOA from controlled chamber open burning of turkey oak (Cubison et al., 2011) …."

[Figure]

**Fig. 4**. Scatter plot of $f_{44}$ vs. $f_{60}$. The gray markers correspond to the measured OA during this study and the Salmon River Complex Fire (SRCF) case study data are colored by cumulative solar radiation ($\sum$SR). In addition, the five OA factors identified in this study are shown as solid squares and the BBOA factors reported in literature where multiple BBOA factors were derived are shown in different markers. The dashed red lines denote $f_{60} = 0.003$ and the boundaries set for BBOA (Ortega et al. 2013). The brown oval encompasses ARCTAS fire plumes sampled above the North America Continent (Cubison et al., 2011).

In terms of comparison with other studies where multiple BBOA factors were derived, we have added the following text at the end of the first paragraph of section 3.3: "… Similarly, previous studies reported the identification of multiple BBOA factors representative of different degree of atmospheric processing (e.g., Bougiatioti et al., 2014; Brito et al., 2014) and varying combustion conditions (e.g., Young et al., 2015; Young et al., 2016). BBOA-1 and BBOA-2 looked more similar to the fresher BBOA factors while BBOA-3 was more similar to the aged BBOA factors

derived in Bougiatioti et al. (2014) and Brito et al. (2014) in terms of mass spectral features (Fig. 4).

Specific comments

Line 152: The 1.5 µg/m3 cutoff is not adequately justified and seeing as many AMS factorisations have been performed successfully with signals lower than this, it would appear to be an odd thing to do. Why was this chosen as the cutoff? What were the actual S/Ns, according to the error model? Why wasn't the low S/N data simply downweighted, as is standard practice? It seems to me that the real benefit of this strategy is so that the factorisation will be of the high-intensity plumes (the subject of interest) than background data (which can hinder convergence not through low S/N but through rotational ambiguity). This is, in its own way, justifiable, but it should be presented as such, rather than a simple S/N issue.

The low loading periods of this study (< 1.5 µg/m$^3$) were also periods not influenced by biomass burning, thus the removal should have little impact on the variance in the BB signals. We initially performed PMF on data from the entire sampling period, but the PMF model didn't converge, due to larger model rotational ambiguity introduced by the data from low loading periods. Downweighting low loading periods didn't allow the model to converge either. We therefore tested different cutoff values of organic mass concentration to retain as many data points as possible while allow PMF to converge and settled at a cutoff point of 1.5 µg m$^{-3}$.

The text has been revised to read: "Periods with organic concentration below 1.5 µg m$^{-3}$ (~ 20% of the total data points), which hindered the model to converge due to increased rotational ambiguity, were excluded from PMF analysis."

Line 207: How was potassium measured? This is important because it affects the credibility of the data. What constitutes 'low'? A quantitative comparison with other studies should be given.

Potassium was measured by examining potassium signal (K$^+$, $m/z$ = 38.963711) in the HR-AMS. The 5-min average detection limit for K$^+$, defined as 3 times the standard deviations (3σ) of the corresponding signals in particle-free ambient air, was estimated at 0.0023 nitrate equivalent µg m$^{-3}$. As shown in the figure below, potassium signals in aerosol were very noisy and the temporal variation of K showed little correlation with BB plumes. To clarify this point, we have added the time series of potassium to the Supplement (Fig. S6).

[Figure]

**Fig. S6**. Time series of $K^+$ measured by the HR-AMS in different chopper positions.

The text has been revised to read: "Note that potassium (K) is frequently used as a tracer for BB aerosol and the presence of K in aerosol particles was clearly observed during high loading periods. However, overall, K concentration in aerosol was very low and noisy throughout this study (Fig. S6), indicating low K contents in wildfire emissions in the western US."

Line 276: While a combined approach to PMF can improve factorisation and aid interpretation, it can also harm the analysis. The fact that the sources and processes governing inorganics are often fundamentally different to those of organics means that their inclusion can introduce 'model error', in turn increasing rotational ambiguity. Also, because of the high 'strength' of the inorganic variables (owing to their relative lack of fragmentation), they may lead the factors, causing the organic data to more reflect inorganic, rather than organic, sources and processes. This note of caution should be added to the text. Did the authors try running PMF without the inorganics? How did the solutions differ?

Although the sources and processes governing inorganics are often different than those of the organics, the same processes that produce secondary inorganic aerosol species also produce SOA. In addition, VOCs are often co-emitted with inorganic aerosol precursors (e.g., $NH_3$, $NO_x$, and $SO_2$). Conducting PMF analysis on the combined spectra of organic and inorganic aerosols is thus technically sound and physically meaningful. Furthermore, the PMF solutions of combined matrix provide information on the distributions of inorganic signals among different factors and the association between inorganic and organic aerosol components in each factor. This information is helpful for interpreting the sources, chemical characteristics, and evolution processes of OA (Sun et al., 2012).

In this study, we performed PMF analyses under various conditions, on the organic matrix only and the combined aerosol matrix for the entire sampling period, BB impacted periods, and clean periods without BB influence, respectively. After extensive evaluation and cross-comparisons of the solutions, we believe that including the inorganic signals in the PMF analysis enable us to better resolve different types of OA for this study. For example, by performing PMF on the organic matrix for "No BB" periods, we observed two types of OOA – an intermediately oxidized OOA associated with boundary layer (BL) dynamics and a highly oxidized one that correlated with sulfate and appeared to represent free tropospheric air masses. When we analyzed the organic matrix of the entire study period, we found that a minimum number of 5 factors is needed to adequately account for the observed variance but the solution resolved only one OOA and four other factors that appeared to represent BBOAs. However, there are indications of splitting and mixing among factors for this solution. The 6-factor solution led to further splitting and mixing of the BBOA factors without being able to resolve two meaningful OOAs. In contrast, performing PMF on the combined inorganic and organic matrix allowed the model to resolve a highly-oxidized background OOA factor associated with ammonium sulfate, an intermediately oxidized background OOA factor driven by BL dynamics, and three distinct BBOA factors for the 5-factor solution.

The following text has been added to the end of the second paragraph of section 2.3: "PMF was also performed on the organic spectra only but wasn't able to resolve two types of OOA (see more detailed discussions in Section 1 of the Supplement)"

In addition, text has been added to the Supplement as Section 1 and reads:

"**Section 1. PMF analysis**

PMF is commonly applied to the organic mass spectral matrix to determine distinct OA factors (Zhang et al., 2011 and references therein), but conducting PMF analysis on the combined spectra of organic and inorganic aerosols allows for the deriving of additional information. In this study, we performed PMF analyses under various conditions, i.e., on organic matrix only and combined aerosol matrix for the entire sampling period, BB impacted periods, and clean periods without BB influence, respectively. PMF analysis on the organic matrix for "No BB" periods resolved two types of oxygenated OA (OOA) – an intermediately oxidized OOA associated with boundary layer (BL) dynamics and a highly oxidized one that correlated with sulfate and appeared to represent free tropospheric air masses. However, PMF analysis on the organic matrix for the entire study period was unsuccessful at retrieving two meaningful OOA factors. A minimum of 5 factors is needed to fully account for the observed variance but the 5-factor solution resolved only one OOA and four other factors that appeared to represent BBOAs. However, there are indications of splitting and mixing among factors for this solution. The 6-factor solution led to further splitting and mixing of the BBOA factors without being able to resolve two meaningful OOAs. In contrast, performing PMF on the combined inorganic and organic matrix allowed the model to resolve a highly-oxidized background OOA factor associated with ammonium sulfate, an intermediately oxidized background OOA factor driven by BL dynamics, and three distinct BBOA factors for the 5-factor solution. In addition, the solutions of the combined matrix provide information on the distributions of inorganic signals among different factors and the association between inorganic and organic aerosol components in each factor. This information is helpful for interpreting the sources, chemical characteristics, and evolution processes of OA (Sun et al., 2012)."

Line 330: The results here do not necessarily prove that BBOA-3 was formed through processing. While the addition of oxygen would be consistent with other observations of ageing in biomass burning, it does not discount the possibility that an amount of it was in the form of primary humic-like substances, which are known to be formed during biomass burning (e.g. http://www.atmos-chem-phys.net/6/5213/2006/). These generally bear a resemblance to highly oxygenated secondary organic aerosol in terms of their volatility, chemical functionality and AMS mass spectra, so a caveat should be added.

According to the reviewer's comment, we have revised the text to: "These results indicate that BBOA-3 was associated with wildfires and likely formed both through rapid processing near the wildfire source and during transport to MBO. However, given that humic-like substances (HULIS) are a known component of BB emissions and that these substances resemble BBOA-3 in terms of AMS mass spectrum, high degree of oxygenation, and low volatility (Dinar et al., 2006; Adler et al., 2011), it is possible that a fraction of BBOA-3 was HULIS as well."

Anonymous Referee #2

General comments:

This manuscript reports HR-AMS measurements of fresh and aged biomass burning emissions observed from the Biomass Burning Observation Project (BBOP) field campaign in summer 2013. PMF analysis and other measurements were performed to investigate atmospheric chemistry of biomass burning organic aerosol (BBOA) in Western US. This study observed that all BBOA factors (BBOA-1, 2 and 3) composed of a larger fraction of high molecular weight organics compared to oxygenated organic aerosols factors (BL-OOA and LV-OOA). Thermodeunder measurements further suggested the presence of low-volatility BBOA in aged biomass burning plume, which is consistent to some recent literature. More importantly, a case study provides insight into the net production of organic aerosol mass due to atmospheric aging of wildfire plume, which is of great interest to the atmospheric aerosol community. The measurements and data analysis were well performed and the major scientific arguments are convincing. The manuscript is well organized and written in general. I recommend this manuscript to be published in Atmospheric Chemistry and Physics after addressing the specific comments below.

Specific comments:

1. PMF analysis, Line 149-156: It is uncommon to run PMF with inorganic components as those peaks could be too strong that drive the overall PMF solution. Therefore, it is recommended to better highlight the merits and rationales behind to include inorganic components in the PMF analysis, and briefly compare their existing PMF results to that without inorganic fragments.

The discussions on the merit of including the inorganic signals in the PMF analysis for this study are given on page 4 of this document, i.e., in our response to reviewer #1's comment starting with "Line 276". Following are the texts added to the revised manuscript and supplementary information:

At the end of the second paragraph of section 2.3 "PMF was also performed on the organic spectra only but wasn't able to resolve two types of oxygenated OA (Section 1 of the Supplement)"

Section 1 in the Supplement: "**Section 1. PMF analysis**
   PMF is commonly applied to the organic mass spectral matrix to determine distinct OA factors (Zhang et al., 2011 and references therein), but conducting PMF analysis on the combined spectra of organic and inorganic aerosols allows for the deriving of additional information. In this study, we performed PMF analyses under various conditions, i.e., on organic matrix only and combined aerosol matrix for the entire sampling period, BB impacted periods, and clean periods without BB influence, respectively. PMF analysis on the organic matrix for "No BB" periods resolved two types of oxygenated OA (OOA) – an intermediately oxidized OOA associated with boundary layer (BL) dynamics and a highly oxidized one that correlated with sulfate and appeared to represent free tropospheric air masses. However, PMF analysis on the organic matrix for the entire study period was unsuccessful at retrieving two meaningful OOA factors. A minimum of 5 factors are needed to fully account for the observed variance but the 5-factor solution resolved only one OOA and four other factors that appeared to represent BBOAs. However, there are indications of splitting and mixing among factors for this solution.

The 6-factor solution led to further splitting and mixing of the BBOA factors without being able to resolve two meaningful OOAs. In contrast, performing PMF on the combined inorganic and organic matrix allowed the model to resolve a highly-oxidized background OOA factor associated with ammonium sulfate, an intermediately oxidized background OOA factor driven by BL dynamics, and three distinct BBOA factors for the 5-factor solution. In addition, the solutions of the combined matrix provide information on the distributions of inorganic signals among different factors and the association between inorganic and organic aerosol components in each factor. This information is helpful for interpreting the sources, chemical characteristics, and evolution processes of OA (Sun et al., 2012)."

2. Potassium detection, line 207-211: The author mentioned that the potassium signal was low throughout the whole period of study but it is more important to examine if the temporal variation of potassium correlates with those of the identified BBOA factors. In addition, potassium background in AMS data is high in general due to surface ionization of tungsten vaporizer. Please report detection limit of potassium and compared to the ambient data.

For this study, the detection limit of potassium, defined as 3 times the standard deviations ($3\sigma$) of the corresponding signals in particle-free ambient air, was 0.0023 nitrate equivalent µg m$^{-3}$. Upon examining the high-resolution spectra, the presence of K during large BB plume periods was evident. However, potassium signals in aerosol were generally low and noisy throughout this study and the temporal variation of K showed little correlation with BB plumes (Fig. S6). A figure of the time series of K$^{+}$ measured by the HR-AMS under different chopper positions have been added to the Supplement (Fig. S6).

[Figure]

**Fig. S6**. Time series of K$^{+}$ measured by the HR-AMS in different chopper positions.

The text has been revised to read: "Note that potassium (K) is frequently used as a tracer for BB aerosol and the presence of K in aerosol particles was clearly observed during high loading periods. However, overall, K concentration was very low and noisy throughout this study (Fig. S6), indicating low K contents in wildfire emissions in the western US."

3. Ammonium level in the BB plume, line 235: Figure S7 shows that the concentrations of ammonium were much higher than that required to completely neutralize sulfate, nitrate and chloride when organic loadings were high (e.g. > 50 ug/m3) due to the presence of biomass burning plume. It is well-known that biomass burning can produce significant amounts of nitrogen-containing organics such as amine. Please discuss if the observed NHx fragments in biomass burning plume were due to the increased level of amine in particle phase.

It is true that biomass burning can emit significant amounts of nitrogen-containing compounds and that these compounds, such as amines, can produce $NH_x^+$ ions in the AMS (Ge et al., 2014). In this study, we indeed observed good correlations between $C_xH_yN^+$ ions and biomass burning tracers (e.g., CO, AMS $C_2H_4O_2^+$, and $C_3H_5O_2^+$), suggesting emissions of amino compounds from wildfires in the western US. However, the low concentrations of the $C_xH_yN^+$ ions (~ 0.3% of total organic signal) indicate only a small, if any, influence of these compounds on $NH_4^+$ quantification. Another possible explanation for the higher concentrations of ammonium relative to those required for neutralizing sulfate, nitrate and chloride was the presence of organic acids, as suggested by the intense signals of $CO_2^+$ ($m/z = 44$) and $CHO_2^+$ ($m/z = 45$) in the HRMS (Fig. 6f). Carboxylic acids were not included in the calculation of anion concentrations.

The third paragraph of section 3.2.1 has been revised accordingly: "…Note that for high organic loading (> 50 µg m$^{-3}$) periods, excess ammonium relative to sulfate, nitrate, and chloride was frequently observed. A possible reason is the presence of significant amounts of organic anions in aerosol. Indeed, $CO_2^+$ ($m/z = 44$) and $CHO_2^+$ ($m/z = 45$) – ion fragments for carboxylic acids – were found to dominate the HRMS of aerosol during periods of high OA loading (Fig. S7f). Another possible reason is overestimation of ammonium concentration. Biomass burning can emit significant amounts of nitrogen-containing organic compounds, including amines. These compounds can produce $NH_x^+$ ions in the AMS, although generally produce significantly more $C_xH_yN^+$ ions (Ge et al., 2014). Tight correlations between $C_xH_yN^+$ ions and biomass burning tracers (e.g., CO, $C_2H_4O_2^+$, and $C_3H_5O_2^+$) were observed, suggesting that amino compounds were likely emitted from wildfires in the western US. However, the low abundance of $C_xH_yN^+$ (~ 0.3% of total organic signal) indicates that organic nitrogen compounds unlikely had a noticeable influence on ammonium quantification during this study. …"

Minor and technical comments:

1. Line 58-60: Please add the recent publication by Gilardoni et al. (2016) that reports SOA formation in the aged biomass burning emission through aqueous-phase processing.

Reference added.

2. Line 158: It is unclear whether time-dependent or average CE applied to the PMF results.

We applied the time-dependent CE the PMF results. We have revised the text to clarify this point: "After PMF analysis, the mass concentration of each OA factor was derived from the sum of organic signals in the corresponding mass spectrum after applying the RIE (= 1.4) for organics and the time-dependent CE determined based on aerosol composition (see previous discussion)."

3. Line 167-169: Uncertainties of the mass fraction remaining (MFR) for each factor are increasing with the operating temperature of thermodenuder, especially for more volatile species. Please highlight the potential uncertainties in the revised version.

In response to the reviewer's comment, the following text has been added to the last paragraph of section 2.3: "…The mass fraction remaining (MFR) of a factor at each TD temperature was then determined as the slope from orthogonal fit between the time series after TD and the ambient time series. Note that the uncertainties of the MFR likely increase with TD temperature, especially for more volatile species, possibly due to changes in particle collection efficiency and decreased concentration (thus lower S/N). Indeed, as shown in Fig. S4, the correlation coefficients between the TD-processed aerosol species and the ambient data decreased with increased TD temperature…"

The error bars shown in Fig. 6g of the revised manuscript represent the standard deviation of the fitted slopes between the TD and the ambient data (i.e., MFR). We have revised the caption of this figure to highlight this information: "…(g) Volatility profiles of OA factors, sulfate, and nitrate, with error bars showing the standard deviation of the calculated slope, i.e., mass fraction remaining."

4. Line 345: It should be "Fig. 4i" instead of "Fig. 5i".

The reviewer was correct. As a new Figure 4 has been added in the revised manuscript, this is figure has now become "Fig. 5i".

**Reference**

Collier, S., Zhou, S., Onasch, T. B., Jaffe, D. A., Kleinman, L., Sedlacek, A. J., Briggs, N. L., Hee, J., Fortner, E., Shilling, J. E., Worsnop, D., Yokelson, R. J., Parworth, C., Ge, X., Xu, J., Butterfield, Z., Chand, D., Dubey, M. K., Pekour, M. S., Springston, S., and Zhang, Q.: Regional Influence of Aerosol Emissions from Wildfires Driven by Combustion Efficiency: Insights from the BBOP Campaign, Environmental Science & Technology, 50, 8613-8622, 2016.

Cubison, M. J., Ortega, A. M., Hayes, P. L., Farmer, D. K., Day, D., Lechner, M. J., Brune, W. H., Apel, E., Diskin, G. S., Fisher, J. A., Fuelberg, H. E., Hecobian, A., Knapp, D. J., Mikoviny, T., Riemer, D., Sachse, G. W., Sessions, W., Weber, R. J., Weinheimer, A. J., Wisthaler, A., and Jimenez, J. L.: Effects of aging on organic aerosol from open biomass burning smoke in aircraft and laboratory studies, Atmospheric Chemistry and Physics, 11, 12049-12064, 2011.

Ge, X., Shaw, S. L., and Zhang, Q.: Toward Understanding Amines and Their Degradation Products from Postcombustion $CO_2$ Capture Processes with Aerosol Mass Spectrometry, Environmental Science & Technology, 48, 5066-5075, 2014.

Jolleys, M. D., Coe, H., McFiggans, G., McMeeking, G. R., Lee, T., Kreidenweis, S. M., Collett, J. L., and Sullivan, A. P.: Organic aerosol emission ratios from the laboratory combustion of biomass fuels, Journal of Geophysical Research: Atmospheres, 119, 2014JD021589, 2014.

Jolleys, M. D., Coe, H., McFiggans, G., Taylor, J. W., O'Shea, S. J., Le Breton, M., Bauguitte, S. J. B., Moller, S., Di Carlo, P., Aruffo, E., Palmer, P. I., Lee, J. D., Percival, C. J., and Gallagher, M. W.: Properties and evolution of biomass burning organic aerosol from Canadian boreal forest fires, Atmos. Chem. Phys., 15, 3077-3095, 2015.

Ortega, A. M., Day, D. A., Cubison, M. J., Brune, W. H., Bon, D., de Gouw, J. A., and Jimenez, J. L.: Secondary organic aerosol formation and primary organic aerosol oxidation from biomass-burning smoke in a flow reactor during FLAME-3, Atmos. Chem. Phys., 13, 11551-11571, 2013.

Sun, Y. L., Zhang, Q., Schwab, J. J., Yang, T., Ng, N. L., and Demerjian, K. L.: Factor analysis of combined organic and inorganic aerosol mass spectra from high resolution aerosol mass spectrometer measurements, Atmospheric Chemistry and Physics, 12, 8537-8551, 2012.

Weimer, S., Alfarra, M. R., Schreiber, D., Mohr, M., Prévôt, A. S. H., and Baltensperger, U.: Organic aerosol mass spectral signatures from wood-burning emissions: Influence of burning conditions and wood type, Journal of Geophysical Research: Atmospheres, 113, D10304, 2008.